# SCALING UP MASKED DIFFUSION MODELS ON TEXT

**Shen Nie**[1,2*]**, Fengqi Zhu**[1,2]**, Chao Du**[3‡]**, Tianyu Pang**[3]**, Qian Liu**[3]**, Guangtao Zeng**[4]
**Min Lin**[3]**, Chongxuan Li**[1,2‡†]

[1]Gaoling School of Artificial Intelligence, Renmin University of China
[2]Beijing Key Laboratory of Big Data Management and Analysis Methods
[3]Sea AI Lab, Singapore    [4]Singapore University of Technology and Design
{nieshen, fengqizhu}@ruc.edu.cn; {duchao, tianyupang, liuqian}@sea.com;
zengguangtao98@gmail.com; linmin@sea.com; chongxuanli@ruc.edu.cn

## ABSTRACT

Masked diffusion models (MDMs) have shown promise in language modeling, yet their scalability and effectiveness in core language tasks, such as text generation and language understanding, remain underexplored. This paper establishes the first scaling law for MDMs, demonstrating a scaling rate comparable to autoregressive models (ARMs) and a relatively small compute gap. Motivated by their scalability, we train a family of MDMs with up to 1.1 billion (B) parameters to systematically evaluate their performance against ARMs of comparable or larger sizes. Fully leveraging the probabilistic formulation of MDMs, we propose a simple yet effective *unsupervised classifier-free guidance* that effectively exploits large-scale unpaired data, boosting performance for conditional inference. In language understanding, the 1.1B MDM outperforms the 1.1B TinyLlama model trained on the same data across four of eight zero-shot benchmarks. Notably, it achieves competitive math reasoning ability with the 7B Llama-2 model on the GSM8K dataset. In text generation, MDMs with 16 times more pre-training time offer a flexible trade-off against ARMs with the accelerated sampling technique KV-Cache: MDMs match ARMs in performance while being 1.4 times faster during sampling. Moreover, MDMs address challenging tasks for ARMs by effectively handling bidirectional reasoning and adapting to temporal shifts in data. Notably, a 1.1B MDM breaks the *reverse curse* encountered by much larger ARMs with significantly more data and computation, such as 13B Llama-2 and 175B GPT-3. Our code is available at https://github.com/ML-GSAI/SMDM.

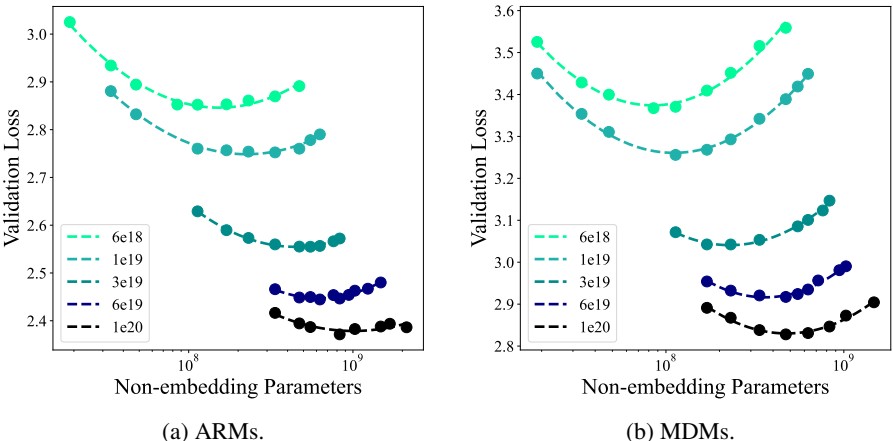

(a) ARMs.             (b) MDMs.

Figure 1: **IsoFLOP curves** plot optimal model sizes under fixed computation budgets. The optimal MDMs validation loss exhibits power-law scaling, decreasing at a rate comparable to that of ARMs.

---

*Work done during Shen Nie's internship at Sea AI Lab.
‡Project leaders. †Correspondence to Chongxuan Li.

# 1 INTRODUCTION

Autoregressive models (ARMs) have long been regarded as the gold standard in probabilistic language modeling. Their ability to predict the next token, grounded in the chain rule, naturally aligns with the sequential nature of language and scales effectively (Radford, 2018; Radford et al., 2019; Brown, 2020; OpenAI, 2022; Achiam et al., 2023; Touvron et al., 2023a;b; Dubey et al., 2024) when integrated with Transformers (Vaswani, 2017). However, ARMs exhibit inherent limitations, particularly in reasoning tasks that require bidirectional context understanding or handling temporal shifts in data. These shortcomings, widely recognized as the *reverse curse* (Berglund et al., 2023) and *temporal quality degradation* (Vela et al., 2022), significantly hinder their applicability in complex language modeling scenarios. Additionally, their linear sampling time growth w.r.t. the output length poses practical challenges for long text generation.

The limitations of ARMs have sparked interest in an alternative approach: masked diffusion models (MDMs) (Austin et al., 2021; Hoogeboom et al., 2021b;a; He et al., 2022; Campbell et al., 2022; Meng et al., 2022; Sun et al., 2022; Lou et al., 2023; Sahoo et al., 2024; Shi et al., 2024; Ou et al., 2024). MDMs present a promising alternative due to their unique probabilistic framework, which enables flexible bidirectional context modeling by filling in masked positions across a sequence. Recent advances (Lou et al., 2023; Sahoo et al., 2024; Shi et al., 2024; Ou et al., 2024) have shown promise in unconditional text generation and zero-shot perplexity evaluation. Despite recent progress, the scalability of MDMs and their effectiveness in critical language tasks, such as conditional generation and language understanding, remain open questions. Furthermore, it is still unclear whether MDMs can address the inherent limitations of ARMs, such as improving bidirectional reasoning capabilities.

Given that scalability and generality across tasks are core attributes of large language models, advancing MDMs requires not only a focus on algorithm design (Austin et al., 2021; Lou et al., 2023; Sahoo et al., 2024; Shi et al., 2024; Ou et al., 2024) but also attention to an orthogonal dimension: the exploration of scalability and generality. From this perspective, this paper challenges the longstanding dominance of ARMs by presenting a comprehensive study of MDMs regarding key factors in language models: scalability, language understanding capabilities, and conditional generation performance. To achieve this, we train a family of MDMs with up to 1.1 billion (B) parameters on a large-scale dataset and establish the first scaling law for MDMs. Leveraging their unique probabilistic framework, we propose a simple yet effective *unsupervised classifier-free guidance (CFG)* mechanism to leverage unsupervised data to enhance inference performance in language tasks involving conditional distributions. Notably, unsupervised CFG does not rely on paired data as standard CFG (Ho & Salimans, 2022) but can still benefit from paired data when available, achieving performance that surpasses standard CFG. Supported by the scaling law and unsupervised CFG, our extensive experiments yield the following key findings:

- **Strong scalability.** As the IsoFLOP analysis (Hoffmann et al., 2022) scaling compute budgets from $6 \times 10^{18}$ to $10^{20}$ FLOPs (see Fig. 1), the optimal validation loss of MDMs decreases according to a power law, with a rate matching that of ARMs (see Fig. 2). While MDMs maintain a constant computation gap of 16 times compared to ARMs, this gap is smaller than the factor of 64 observed in continuous diffusion models (Gulrajani & Hashimoto, 2024) and can be further minimized with future optimizations.
- **Competitive in language understanding.** Across eight standard zero-shot benchmarks, including tasks like *commonsense reasoning* and *reading comprehension*, our 1.1B MDM outperforms the larger 1.5B GPT-2 model on six tasks and the same-sized TinyLlama (with equivalent pre-training FLOPs) on four tasks. Moreover, the 1.1B MDM demonstrates competitive *math reasoning* performance compared to the 7B Llama-2 on the GSM8K dataset, while utilizing less than $5\%$ of its pre-training FLOPs.
- **Flexible trade-off in conditional generation.** On the standard MT-Bench, a 1.1B MDM matches the performance of a same-sized ARM while achieving a 1.4 times speedup in sampling time. By increasing sampling steps, MDMs can further improve generation quality at the cost of being 1.4 times slower. Notably, ARMs are equipped with KV-cache, a technique to speed up sequential sampling while MDMs exploit no system optimization but require 16 times pre-training time.
- **Addressing challenging tasks for ARMs.** MDMs effectively relieve *temporal quality degradation* (Vela et al., 2022) compared to a same-sized ARM and successfully overcome the *reverse*

*curse* (Berglund et al., 2023) encountered by much larger ARMs with significantly more data and computation, such as 13B Llama-2 and 175B GPT-3.

## 2 MASKED DIFFUSION MODELS ON TEXT

In analogy to continuous diffusion models (Sohl-Dickstein et al., 2015; Ho et al., 2020; Song et al., 2020), MDMs (Austin et al., 2021; Lou et al., 2023; Ou et al., 2024) also introduce a forward process that gradually adds noise to the data and learn a corresponding reverse process to generate samples. Our basic approach is built upon Ou et al. (2024), an advanced MDM suitable for scaling.

**Forward process.** Let $K$ and $L$ denote the vocabulary size and sentence length respectively. Given a sentence $\boldsymbol{x}_0 \in \{0, 1, \ldots, K-1\}^L$ and a noise level $t \in [0, 1]$, the forward process in MDMs randomly and independently masks out tokens in the sentence, formulated as follows:

$$q_{t|0}(\boldsymbol{x}_t|\boldsymbol{x}_0) = \prod_{i=0}^{L-1} q_{t|0}(\boldsymbol{x}_t^i|\boldsymbol{x}_0^i) \quad \text{and} \quad q_{t|0}(\boldsymbol{x}_t^i|\boldsymbol{x}_0^i) = \begin{cases} \alpha_t, & \boldsymbol{x}_t^i = \boldsymbol{x}_0^i, \\ 1 - \alpha_t, & \boldsymbol{x}_t^i = m, \end{cases} \tag{1}$$

where $\boldsymbol{x}^i$ denotes the $i$-th element of $\boldsymbol{x}$, $m$ denotes the mask token (Devlin, 2018), $\boldsymbol{x}_t$ denotes the noisy data at time $t$ and $q_0(\cdot)$ is the data distribution $p_{\text{data}}(\cdot)$. We set the hyperparameter $\alpha_t$ as $1 - t$ for the best empirical performance as suggested in previous work (Lou et al., 2023; Sahoo et al., 2024; Shi et al., 2024).

**Reverse process.** The reverse process in MDMs iteratively recover values for masked tokens, starting from a mask sequence $\boldsymbol{x}_1$. Let $0 \leq s < t \leq 1$, the reverse process is characterized by

$$q_{s|t}(\boldsymbol{x}_s|\boldsymbol{x}_t) = \prod_{i=0}^{L-1} q_{s|t}(\boldsymbol{x}_s^i|\boldsymbol{x}_t) \text{ and } q_{s|t}(\boldsymbol{x}_s^i|\boldsymbol{x}_t) = \begin{cases} 1, & \boldsymbol{x}_t^i \neq m, \boldsymbol{x}_s^i = \boldsymbol{x}_t^i, \\ \frac{s}{t}, & \boldsymbol{x}_t^i = m, \boldsymbol{x}_s^i = m, \\ \frac{t-s}{t} q_{0|t}(\boldsymbol{x}_s^i|\boldsymbol{x}_t), & \boldsymbol{x}_t^i = m, \boldsymbol{x}_s^i \neq m, \\ 0, & \text{otherwise}. \end{cases} \tag{2}$$

Here $q_{0|t}(\cdot|\cdot)$ is the data prediction model (Ho et al., 2020) to be learned. Notably, Ou et al. (2024) revealed an intrinsic property of MDMs that $q_{0|t}(\cdot|\cdot)$ can be represented by conditional distributions on clean data $p_{\text{data}}(\cdot|\cdot)$ independently from the time $t$, distinct from other diffusion. Formally,

$$q_{0|t}(\boldsymbol{x}_0^i|\boldsymbol{x}_t) = p_{\text{data}}(\boldsymbol{x}_0^i|\boldsymbol{x}_t^{\text{UM}}), \tag{3}$$

where $\boldsymbol{x}_t^{\text{UM}}$ collects all unmasked tokens in noisy data $\boldsymbol{x}_t$ and $p_{\text{data}}(\cdot|\cdot)$ is irrelevant to $t$.[1]

**Training objective.** A distribution $p_{\boldsymbol{\theta}}(\boldsymbol{x}_0^i|\boldsymbol{x}_t)$ parameterized by $\boldsymbol{\theta}$ is employed to approximate $p_{\text{data}}(\boldsymbol{x}_0^i|\boldsymbol{x}_t^{\text{UM}})$, optimizing the following upper bound on negative log-likelihood (Ou et al., 2024):

$$-\log p_{\boldsymbol{\theta}}(\boldsymbol{x}_0) \leq \int_0^1 \frac{1}{t} \mathbb{E}_{q(\boldsymbol{x}_t|\boldsymbol{x}_0)} \left[ \sum_{\{i|\boldsymbol{x}_t^i=m\}} -\log p_{\boldsymbol{\theta}}(\boldsymbol{x}_0^i|\boldsymbol{x}_t) \right] dt \triangleq \mathcal{L}. \tag{4}$$

We emphasize that the formulation is particularly suitable for scaling. First, it is among the best MDMs w.r.t. zero-shot perplexity (Ou et al., 2024). Second, it removes the timestep from input and minimally modifies the original Transformers (see Sec. 3). Third, it enables unsupervised classifier-free guidance, which does not rely on paired data yet is effective in language tasks (see Sec. 4).

## 3 SCALING LAWS FOR MASKED DIFFUSION MODELS

Scaling laws (Kaplan et al., 2020; Hoffmann et al., 2022) characterize the quantitative power-law relationship between model performance and computational resources under constraints, significantly influencing the progress of large ARMs. Previous work Ye et al. (2023) fine-tunes pre-trained XLM-RoBERTa (Goyal et al., 2021; Conneau, 2019) models into MDMs and investigates scaling trends by varying the size of XLM-RoBERTa. However, a detailed exploration of scaling laws for MDMs,

---

[1]For example, if $\boldsymbol{x}_t = [3, 5, m, 2]$, then $\boldsymbol{x}_t^{\text{UM}} = [3, 5, \cdot, 2]$ and $p_{\text{data}}(\cdot|[3, 5, \cdot, 2])$ is irrelevant to $t$.

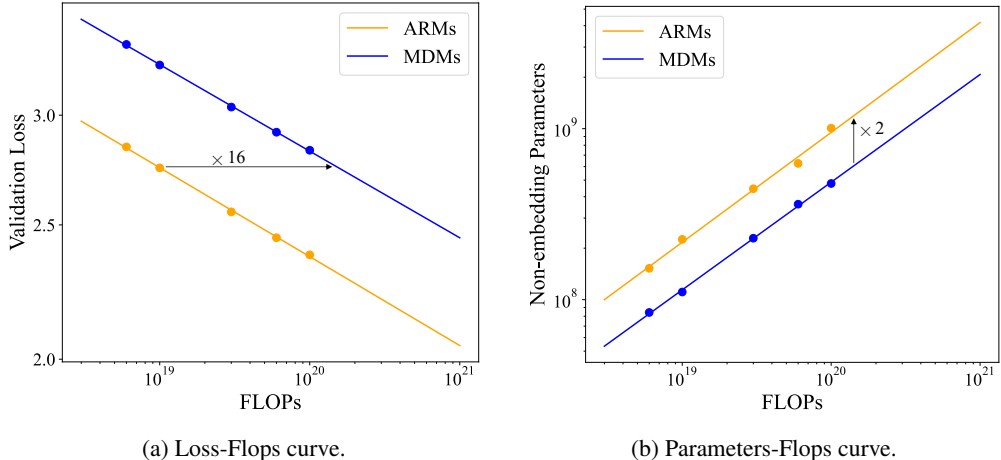

(a) Loss-Flops curve.  (b) Parameters-Flops curve.

Figure 2: **Scaling laws for MDMs.** Compared to ARMs, MDMs demonstrate competitive scalability with comparable scaling rates and similar scaling behavior on utilizing the parameter capacity.

along with a fair comparison to ARMs in terms of scalability, remains absent. In this section, we address these two key questions. Our results reveal the strong scalability of MDMs, highlighting their potential as a competitive alternative to ARMs in language modeling.

**Model.** We employ a Transformer decoder for ARMs and the corresponding Transformer encoder for MDMs (note that it is unnecessary to input timestep $t$ according to Eq. (3)). The differences between these architectures are: (1) the encoder has an additional dimension in its embedding layer for the mask token, and (2) the encoder's self-attention does not use a causal mask. All other architectural settings (e.g., depth, hidden size, and number of heads) remain consistent in both models.

We further enhance both models with several techniques inspired by advanced language models like Llama (Touvron et al., 2023a;b). Specifically, we adopt Pre-LayerNorm with RMSNorm (Zhang & Sennrich, 2019) for better stability, use SwiGLU (Shazeer, 2020) as the activation function to enhance non-linearity, and implement RoPE (Su et al., 2024) for more expressive positional encoding.

**Data.** The well-known Chinchilla scaling law (Hoffmann et al., 2022) utilizes a large dataset with more data than the number of training tokens. Motivated by it, we employ the open-source SlimPajama dataset (Soboleva et al., 2023), a multi-corpora dataset comprising 627 billion tokens, which is sufficiently large for all of our experiments. For simplicity and fairness, we employ the Llama-2 tokenizer (Touvron et al., 2023b) for both ARMs and MDMs. Additionally, we set the context length to 2048. Further implementation details are provided in Appendix B.2.

**IsoFLOP analysis.** We conduct a standard IsoFLOP analysis (Hoffmann et al., 2022) to identify the optimal allocation between the non-embedding parameters $N$ and dataset size $D$. Specifically, building on prior studies (Kaplan et al., 2020; Hoffmann et al., 2022), we scale the compute budget $C$ from $6 \times 10^{18}$ to $10^{20}$ FLOPs. For a fixed $C$, we vary $N$ and $D$ such that $C = 6ND$, a relationship valid for both ARMs and MDMs. We fit a quadratic function to capture the relationship between the validation loss $\mathcal{L}$ and the logarithm of the parameter size $\log N$. Specifically, the loss function $\mathcal{L}$ of MDMs is defined in Eq. (4). This regression allows us to determine the optimal parameter size $N_C^*$, which corresponds to the minimum validation loss $\mathcal{L}_C^*$ for a given compute budget $C$. The IsoFLOP analysis results are visualized in Fig. 1.

**Scaling laws.** After obtaining the optimal validation losses for the corresponding compute budget in $\{C_0, C_1, \ldots, C_{n-1}\}$, we fit the following scaling law to model the relationship between them:

$$\min_{\alpha, \beta} \sum_{i=0}^{n-1} \left( \log \mathcal{L}_{C_i}^* - \alpha \log C_i - \beta \right)^2. \tag{5}$$

Let $\alpha^*$ and $\beta^*$ denote the solution of Eq. (5) and the validation loss empirically follows $\mathcal{L} = e^{\beta^*} C^{\alpha^*}$.

As illustrated in Fig. 2a, the validation loss of MDMs decreases according to a power law as the compute budget increases, following a rate similar to that of ARMs. However, MDMs still require approximately 16 times more computational resources than ARMs to achieve comparable validation losses. Furthermore, the optimal model size also follows a power-law relationship with the compute budget, as shown in Fig. 2b. The optimal size of MDMs is approximately half that of ARMs across different computations, reflecting a similar scaling behavior on utilizing the parameter capacity.

Despite the constant gap in computational resources (i.e., MDMs require 16 times more resources than ARMs), there is still potential to narrow this gap since optimizations for MDMs in model, data, and system remain unexplored. Besides, for reference, Gulrajani & Hashimoto (2024) reported that continuous diffusion models (CDMs) require 64 times more computational resources than ARMs. In Sec. 5, we further emphasize that MDMs achieve competitive results on commonsense reasoning and reading comprehension tasks compared to ARMs under equivalent computational resources. Additionally, our findings show that MDMs exhibit competitive mathematical reasoning capabilities (see Sec. 5) and superior bidirectional modeling ability (see Sec 7.1) when compared to significantly larger ARMs operating with far greater compute budgets.

In conclusion, the comparable scaling rates and the relatively small constant factors suggest that MDMs have strong scalability and promising potential as an alternative to ARMs on a large scale.

## 4 UNSUPERVISED CLASSIFIER-FREE GUIDANCE

We propose a surprisingly simple yet effective approach that leverages unlabeled data to boost performance in various language tasks, dubbed *unsupervised classifier-free guidance (CFG)*.

**CFG.** CFG (Ho & Salimans, 2022) is an effective and versatile technique widely used in both continuous and discrete diffusion models, with applications spanning image (Ho & Salimans, 2022; Chang et al., 2023) and text generation (Lovelace et al., 2024). Rooted in Bayes' rule, CFG simultaneously trains a conditional and an unconditional diffusion model, introducing a rescaled distribution for inference. Specifically, at a given timestep $t \in [0, 1]$, CFG (Chang et al., 2023) is defined as:

$$\tilde{p}_{\boldsymbol{\theta}}(\boldsymbol{x}_0|\boldsymbol{c}, \boldsymbol{x}_t) \propto \frac{p_{\boldsymbol{\theta}}(\boldsymbol{x}_0|\boldsymbol{c}, \boldsymbol{x}_t)^{1+w}}{p_{\boldsymbol{\theta}}(\boldsymbol{x}_0|\boldsymbol{x}_t)^w}, \tag{6}$$

where $\boldsymbol{c}$ is the condition, $w$ is a hyperparameter that flexibly controls the strength of $\boldsymbol{c}$, and $p_{\boldsymbol{\theta}}(\boldsymbol{x}_0|\boldsymbol{c}, \boldsymbol{x}_t)$ and $p_{\boldsymbol{\theta}}(\boldsymbol{x}_0|\boldsymbol{x}_t)$ are the conditional and unconditional models respectively.

Notably, it seems that the conditional model must be trained on paired data before applying CFG. Consequently, to the best of our knowledge, all existing work (Ho & Salimans, 2022; Chang et al., 2023; Lovelace et al., 2024) fall into supervised settings, where paired data are readily available.

**Unsupervised CFG.** We extend CFG to an unsupervised setting by introducing a new formulation:

$$\tilde{p}_{\boldsymbol{\theta}}(\boldsymbol{x}_0|\boldsymbol{c}, \boldsymbol{x}_t) \propto \frac{p_{\boldsymbol{\theta}}(\boldsymbol{x}_0|\boldsymbol{c}, \boldsymbol{x}_t)^{1+w}}{p_{\boldsymbol{\theta}}(\boldsymbol{x}_0|\boldsymbol{m}, \boldsymbol{x}_t)^w}, \tag{7}$$

where $\boldsymbol{m}$ is a mask sequence of the same length as $\boldsymbol{c}$. Compared to Eq. (6), the dummy variable $\boldsymbol{m}$ translates the unconditional distribution to a conditional format without adding new information. For simplicity, we continue to refer to $p_{\boldsymbol{\theta}}(\boldsymbol{x}_0|\boldsymbol{m}, \boldsymbol{x}_t)$ as the unconditional distribution in unsupervised CFG throughout this paper.

The core insight is that an MDM already characterizes both distributions employed in Eq. (7) during unsupervised pretraining. Specifically, in language tasks, both $\boldsymbol{c}$ and $\boldsymbol{x}$ can be viewed as segments of a whole sequence, following the same distribution of unsupervised samples for pretraining.[2] After the pretraining on large-scale text data, MDMs can capture the joint distribution of the whole sequence, i.e., $p_{\text{data}}(\boldsymbol{c}, \boldsymbol{x})$. Under the formulation, MDMs simultaneously learn all conditional distributions on clean data induced by $p_{\text{data}}(\boldsymbol{c}, \boldsymbol{x})$ according to Eq. (3). In particular, we have:

$$p_{\boldsymbol{\theta}}(\boldsymbol{x}_0|\boldsymbol{c}, \boldsymbol{x}_t) \approx p_{\text{data}}(\boldsymbol{x}_0|\boldsymbol{c}, \boldsymbol{x}_t^{\text{UM}}) \quad \text{and} \quad p_{\boldsymbol{\theta}}(\boldsymbol{x}_0|\boldsymbol{m}, \boldsymbol{x}_t) \approx p_{\text{data}}(\boldsymbol{x}_0|\boldsymbol{x}_t^{\text{UM}}), \tag{8}$$

---

[2]E.g., the question "where does the sun rise?" and answer "from the east." is a paired sample but their concatenation "where does the sun rise? from the east." can be modeled by an MDM with unsupervised training.

Table 1: **Ablation of unsupervised CFG without paired data.** Unsupervised CFG significantly improves the zero-shot performance of MDMs across eight tasks.

|  | ARC-e | BoolQ | Hellaswag | Obqa | PIQA | RACE | SIQA | LAMBADA |
|---|---|---|---|---|---|---|---|---|
| MDM w/o CFG | 37.42 | 61.50 | 33.46 | 27.00 | 60.34 | 29.28 | 36.95 | 36.00 |
| MDM w/ CFG | **39.02** | **62.17** | **34.10** | **34.20** | **60.39** | **30.81** | **37.41** | **40.99** |

where both distributions are factorized as in Eq. (3), and the approximation error is due to the gap between the model distribution and the true data distribution. Notably, Eq. (8) also implies that the unconditional distribution $p_{\boldsymbol{\theta}}(\boldsymbol{x}_0|\boldsymbol{x}_t)$ used in standard CFG and the conditional distribution with a dummy variable $p_{\boldsymbol{\theta}}(\boldsymbol{x}_0|\boldsymbol{m}, \boldsymbol{x}_t)$ share a similar role.

We have explained why unsupervised CFG works without paired data (see Sec. 5). Moreover, when paired data are available for downstream tasks, simply fine-tuning the conditional distribution in MDMs—similar to the classical approach used for ARMs—not only further improves the performance of unsupervised CFG but also outperforms the standard CFG trained on paired data, demonstrating its superior capability in leveraging large-scale unpaired data (see Sec. 6). While prior studies Zhao et al. (2021); Holtzman et al. (2021) emphasize the role of unconditional distributions in large language models, we show that unsupervised CFG employs a distinct motivation and formulation, as detailed in Appendix C.1.

## 5 LANGUAGE UNDERSTANDING

We investigate the capabilities of MDMs in language understanding, a critical skill for language models that has been largely overlooked in prior studies (Austin et al., 2021; Lou et al., 2023; Sahoo et al., 2024; Shi et al., 2024; Ou et al., 2024; Gat et al., 2024). Our results show that MDMs are highly competitive to ARMs of similar model sizes and computations.

**Benchmarks.** To provide a comprehensive evaluation, we assess MDMs on eight widely used benchmarks in the zero-shot setting, covering tasks in *commonsense reasoning* and *reading comprehension*: Hellaswag (Zellers et al., 2019), ARC-e (Clark et al., 2018), BoolQ (Clark et al., 2019), PIQA (Bisk et al., 2020), SIQA (Sap et al., 2019), Obqa (Mihaylov et al., 2018), RACE (Lai et al., 2017), and LAMBADA (Paperno et al., 2016). Additionally, we also assess the *math reasoning* ability on the GSM8K (Cobbe et al., 2021a) dataset. Following previous works Ye et al. (2024); Gong et al. (2024), we finetune MDM on the augmented training data (Deng et al., 2023) and test on GSM8K. For a detailed description of these benchmarks, please see Appendix D.

On certain challenging benchmarks such as ARC-c (Clark et al., 2018), WinoGrande (Sakaguchi et al., 2021), and MMLU (Hendrycks et al., 2020), both ARMs and MDMs pre-trained in Sec. 3 perform similarly to random guessing. This is consistent with findings from Wei et al. (2022a), which showed that only ARMs with more than $10^{22}$ training FLOPs can surpass random guessing on MMLU, a phenomenon known as the emergence of new capabilities in large language models. We leave the exploration of their potential emergent abilities at a larger scale as future work.

**Evaluation.** We follow the widely used Language Model Evaluation Harness framework (Gao et al., 2024) to evaluate both ARMs and MDMs. For LAMBADA and GSM8K, given a prompt, we use greedy sampling to generate responses from each model and calculate matching accuracy against the ground truth (see Appendix A for details on the greedy sampling algorithm used for MDMs). For the other tasks, we report the accuracy of each model that selects the correct answer from the provided options based on the given context. Specifically, we compute the likelihood of each option given the prompt and choose the answer with the highest likelihood.

**Fixing the train-test discrepancy.** Due to employing a bidirectional Transformer encoder, MDMs face a train-test discrepancy in context lengths, negatively impacting model performance. Specifically, the training context length is fixed at 2048 tokens, while the testing context length is variable and often shorter. To address this issue, we propose two mitigation strategies: (1) allocate a portion of training data with variable sequence lengths $L \sim \mathcal{U}[1, 2048]$, where $\mathcal{U}[\cdot]$ denotes the uniform distribution; (2) pad sentences with mask tokens to reach 2048 tokens during evaluation.

Table 2: **Evaluation of our 1.1B MDM.** Both the MDM and Llama-2 models are fine-tuned for GSM8K, with all other benchmarks assessed in zero-shot settings. Result marked $^*$ is from Gong et al. (2024). The pre-training datasets consist of approximately 540B tokens for TinyLlama and MDM, compared to 2T tokens for Llama-2. Our 1.1B MDM outperforms the same-size TinyLlama on four out of eight tasks and surpasses the larger GPT-2 on six tasks. Notably, MDM achieves GSM8K accuracy comparable to that of Llama-2, requiring less than 5% of its pre-training FLOPs.

| | FLOPs | ARC-e | BoolQ | Hellaswag | Obqa | PIQA | RACE | SIQA | LAMBADA | GSM8K |
|---|---|---|---|---|---|---|---|---|---|---|
| GPT-2 (1.5B) | - | 51.05 | 61.77 | 50.89 | 32.00 | **70.51** | 33.11 | 40.28 | 44.61 | - |
| TinyLlama (1.1B) | $3.3 \times 10^{21}$ | **52.19** | 59.39 | **54.07** | 33.20 | 70.29 | **35.60** | 39.41 | 43.22 | - |
| MDM (1.1B) | $3.3 \times 10^{21}$ | 48.74 | **62.17** | 51.83 | **33.40** | 69.53 | **35.60** | **41.04** | **52.73** | 58.5 |
| Llama-2 (7B) | $7.8 \times 10^{22}$ | 74.49 | 77.68 | 75.98 | 44.20 | 79.00 | 39.52 | 46.11 | 68.00 | 58.6* |

As present in Appendix C.2, both strategies effectively reduce the train-test discrepancy, and only a small proportion (e.g., 1%) of variable-length training data is sufficient to activate the capability to handle variable length inputs. Given its superior inference efficiency (e.g., 20 times faster than method (2) on the Hellaswag dataset), we employ method (1) in subsequent experiments.

**Flexible likelihood evaluation.** As detailed in Sec. 2, the MDMs model the conditional distribution of clean data, which enables flexible likelihood evaluation. Given a prompt and a sentence $x_0$ of length $L$, we can determine the conditional likelihood using the following methods: (1) employ Monte Carlo estimation to establish a lower bound of the log-likelihood based on Eq. (4); (2) utilize the chain rule to compute the likelihood as $\log p_{\theta}(x_0|\text{prompt}) = \sum_{i=0}^{L-1} \log p_{\theta}(x_0^i|\text{prompt}, x_0^{<i}, m)$.

We observed that the chain rule for likelihood evaluation results in higher accuracy for Obqa and PIQA, while Monte Carlo estimation yields better accuracy for ARC-e, Hellaswag, RACE, and SIQA. Since the answer length of BoolQ consists of only one token ("Yes" or "No"), both methods produce identical results. We adopted this optimal configuration in subsequent experiments and please refer to Appendix C.2 for more details and an empirical explanation.

**Effectivenes of unsupervised CFG without paired data.** In this section, we use a default MDM model with 220M parameters and a training budget of $10^{20}$ FLOPs for efficiency. For likelihood evaluation, we use the rescaled conditional distribution defined in Eq. (7) of unsupervised CFG. Since no paired data is available, standard CFG cannot be applied in this scenario. As shown in Table 1, unsupervised CFG significantly enhances the performance of MDMs across all eight widely used benchmarks, demonstrating its strong capability to leverage unpaired data effectively.

**Competitive zero-shot language understanding performance.** As shown in Table 2, our 1.1B MDM outperforms the 1.1B TinyLlama (Zhang et al., 2024) on four out of eight tasks when trained on the same SlimPajama dataset with an equivalent pre-training FLOPs[3]. Additionally, the 1.1B MDM surpasses the larger 1.5B GPT-2 model on six of the eight benchmarks.

Notably, Table 2 also highlights that our 1.1B MDM achieves performance comparable to the much larger 7B Llama-2 (Touvron et al., 2023b) on mathematical reasoning tasks, as measured by GSM8K accuracy, while requiring less than 5% of the pre-training FLOPs. These findings emphasize the competitive performance of MDMs relative to ARMs.

We also analyze the scaling behavior of MDMs on the language understanding tasks and observe a clear trend: as validation loss decreases, performance on most tasks improves correspondingly. This indicates a positive scaling signal, suggesting that MDMs have the potential to achieve even stronger capabilities with further scaling. Detailed results and analyses are provided in Appendix C.2.

## 6 CONDITIONAL LANGUAGE GENERATION

We investigate the capabilities of MDMs in conditional generation, another core language task largely unexplored previously. Our results show that a 1.1B MDM achieves a more flexible and effective quality-efficiency trade-off during inference than a same-sized ARM that utilizes KV cache.

---

[3]We used the intermediate checkpoint officially provided by TinyLlama. See Table 7 for download links.

Table 3: **Ablation of unsupervised CFG.** The symbols * and † indicate the standard CFG and unsupervised CFG respectively. We report the results with the optimal scale searched in $\{0.4, 0.6, 0.8, 1\}$ for both CFG approaches.

|  | w/o CFG | w/ CFG* | w/ CFG† |
|---|---|---|---|
| Score ↑ | 1.32 | 1.53 | **1.60** |

Table 4: **Conditional generation results.** MDM utilizes 16 times the pre-training time of ARM while ARM utilizes KV-cache.

|  | MDM | | | ARM |
|---|---|---|---|---|
| Score ↑ | 1.40 | 1.56 | 1.60 | 1.57 |
| NFEs ↓ | 64 | 128 | 256 | 325.94 |
| Time ↓ | 204s | 396s | 780s | 555s |

**Evaluation.** Previous studies (Lou et al., 2023; Sahoo et al., 2024; Shi et al., 2024; Ou et al., 2024; Gat et al., 2024) have commonly employed generative perplexity as a metric to assess unconditional generation quality. However, recent work (Zheng et al., 2024) demonstrated that even low-quality samples can yield high generative perplexity scores, suggesting that this metric may not reliably reflect generative quality. Moreover, conditional generation is more widely applicable in real-world scenarios than unconditional generation. Therefore, this paper focuses on conditional generation.

In particular, we employ MT-Bench (Zheng et al., 2023), which uses a strong language model (i.e., GPT-4o (Achiam et al., 2023)) as a judge to score models on open-ended questions. This metric aligns well with human preferences and has become a standard for evaluating large language models.

**Supervised fine-tuning.** We employ an ARM and an MDM, both pre-trained as described in Sec. 3 with 1.1B parameters each. For a meaningful comparison, we evaluate their inference performance and, guided by the scaling law, extend the MDM's pre-training time by a factor of 16. Results using equal computation budgets are provided in Appendix C.3. Following a standard process in language models, we fine-tune both models on the ShareGPT dataset[4], a high-quality dialogue corpus containing user prompts and corresponding ChatGPT responses (OpenAI, 2022).

Since ShareGPT samples vary in length, we pad each sample with the |EOS| token to the maximum sequence length within a batch for the MDM. Following the same approach as for ARMs, we mask the loss on prompts, adding noise only to the response tokens (including the padding |EOS|), while keeping the prompts unchanged in the forward process. As a result, the MDM only tunes the conditional distribution of the response given prompt. We set the sequence length to 1024 and remove the |EOS| token from the generated outputs during inference. For the ARM, generation stops when the |EOS| token is produced, with a maximum sequence length set to 1024 (Zheng et al., 2023). For a fair comparison, we use identical optimizer settings for both models and train for 3 epochs as specified in Zheng et al. (2023). Additional training details are provided in Appendix B.4.

**Effectiveness of unsupervised CFG against standard CFG.** As shown in Table 3, we evaluate the effectiveness of unsupervised CFG by comparing it against several baselines detailed in Appendix B.4. The first one fine-tunes only the conditional distribution of MDM on paired data and sampling without CFG. The second one fine-tunes both conditional and unconditional distributions on paired data and gets samples as in the standard CFG. Additionally, we enhance unsupervised CFG by fine-tuning its conditional distribution on paired data. This is because unsupervised CFG already leverages large-scale pre-trained data to obtain a strong unconditional model. Notably, our unsupervised CFG outperforms the standard CFG, demonstrating its superior ability to leverage large-scale unpaired data considering the paired data for fine-tuning are often of a small scale. For a comprehensive comparison, we also demonstrate that unsupervised CFG outperforms sampling without CFG with half the sampling steps (i.e., equal sampling computation) in Appendix C.3.

**Better efficiency quality trade-off.** We further compare MDMs and ARMs regarding sample quality and efficiency. Our study significantly extends prior work (Lou et al., 2023; Sahoo et al., 2024; Shi et al., 2024; Ou et al., 2024; Gat et al., 2024) in two key aspects: (1) we focus on the more practical and challenging task of conditional generation rather than unconditional generation, and (2) we measure the running time instead of the NFEs, even when ARMs are equipped with the KV-cache, a technique that accelerates sampling by caching intermediate features during sequential generation.

---

[4] https://sharegpt.com/

Table 5: **Results on breaking the reverse curse.** The performance of GPT-3 and Llama-2 is sourced from Berglund et al. (2023) and Lv et al. (2023), respectively. All models are fine-tuned on the same dataset for 10 epochs. For MDM, we use a CFG scale of 0.8. While ARMs and T5 struggle to handle reverse queries, MDMs effectively overcome the reverse curse and maintain performance in the same direction.

| | DescriptionToName | | NameToDescription | | | |
| | Same direction | Reverse direction | Same direction | | Reverse direction | |
| | Acc. ↑ | Acc. ↑ | Acc. ↑ | BLEU ↑ | Acc. ↑ | BLEU ↑ |
|---|---|---|---|---|---|---|
| GPT3 (175B) | 97 | 0 | **50** | - | 0 | - |
| Llama-2 (13B) | 99 | 0 | - | 74 | - | 19 |
| T5 (3B) | **100** | 0 | 47 | **87** | 0 | 20 |
| MDM (1.1B) | 97 | **92** | 49 | 76 | **37** | **67** |

Built upon the unsupervised CFG, MDMs demonstrate a more flexible and effective trade-off between efficiency and quality in conditional generation compared to ARMs. As shown in Table 4, a 1.1B MDM matches the performance of a similarly sized ARM while achieving a 1.4 times speedup in sampling time. Conversely, by increasing the number of sampling steps (at the cost of being 1.4 times slower), MDMs can surpass ARMs in generation quality. All experiments in Table 4 are conducted on a single NVIDIA A100-40GB GPU. These results indicate that MDMs hold promise for conditional generation tasks, such as chat-based applications, where the ability to balance speed and quality is critical.

## 7 CHALLENGING TASKS FOR ARMS

We demonstrate that MDMs exhibit distinct advantages over ARMs in tackling two critical challenges: *reverse curse* (Berglund et al., 2023) and *temporal quality degradation* (Vela et al., 2022).

### 7.1 BREAKING THE REVERSE CURSE

Berglund et al. (2023) introduced the concept of the reverse curse, which refers to the difficulty of ARMs in generalizing bidirectional relationships. Specifically, this occurs when a model is trained on information in the form "A is B" but fails to infer the reverse relationship "B is A." For example, a model trained on the fact "Valentina Tereshkova was the first woman to travel to space" may not correctly answer the reverse question "Who was the first woman to travel to space?" This limitation raises concerns about whether large language models genuinely possess logical reasoning capabilities (Berglund et al., 2023).

**Setup.** We evaluate MDMs on the same reverse curse dataset used by Berglund et al. (2023), which consists of fictitious statements in the format "⟨name⟩ is ⟨description⟩" and the reversals. We fine-tune MDMs on these statements and assess their performance using questions not seen during training. To ensure a comprehensive comparison, we additionally fine-tuned the T5 (Raffel et al., 2020) model using the same dataset (see Appendix B.5 for details). Following the same protocol as Berglund et al. (2023), we generate responses via greedy sampling and report the exact match accuracy. Additionally, we use the BLEU metric (Papineni et al., 2002) to evaluate the quality of name-to-description generation, as suggested by Lv et al. (2023).

**Results.** As shown in Table 5, both the T5 (Raffel et al., 2020) model and advanced ARMs achieve zero accuracy and low BLEU scores when prompted with reverse queries. In contrast, MDMs achieve substantially higher scores across both metrics, despite using significantly fewer parameters, less computation, and a smaller training dataset. Specifically, our MDM uses only 10% parameters, 1% computation, and 10% data compared to 13B Llama-2. Besides, MDMs perform similarly to ARMs with queries in the same direction. These results indicate the power of MDMs in capturing bidirectional relationships and logical structures. This capability arises from the training objective of MDMs (i.e., Eq. (4)), designed to model all conditional distributions within the data. Further, we

Table 6: **Perplexity (↓) results on relieving temporal quality degradation.** The symbol * indicates the training dataset. MDM demonstrates superior robustness to temporal shifts than ARM.

| | SlimPajama* (before Jun. 2023) | Fineweb (Feb. & Mar. 2024) | Fineweb (Apr. 2024) |
|---|---|---|---|
| ARM | **17.34** | 27.01 | 26.93 |
| MDM | 18.02 | **24.06** | **24.01** |

provide additional clarification on Table 5 (e.g., the lower accuracy of MDM in the reverse direction compared to the same direction) in Appendix C.4.

## 7.2 RELIEVING THE TEMPORAL QUALITY DEGRADATION

Vela et al. (2022) highlight a common and challenging issue for modern AI models, including language models: model performance is sensitive to the temporal alignment between the training and test data, particularly when new data fall outside the temporal scope of the training set.

**Setup.** To evaluate the impact of temporal shifts, we train both ARMs and MDMs on the SlimPajama dataset (Soboleva et al., 2023) (see Sec. 3), released in 2023, and test them on the FineWeb dataset (Penedo et al., 2024), which contains samples from February&March, and April of 2024. We extract the first 0.5 billion tokens from each period for evaluation. We use models of equal size (220M parameters) that achieve similar validation losses on SlimPajama. However, it is worth noting that MDMs require 16 times more computation to reach this performance level.

**Results.** As shown in Table 6, although the MDM achieves slightly higher perplexity on the standard validation set (i.e., SlimPajama), it outperforms the ARM on the newer 2024 data. While the exact mechanism remains unclear, we hypothesize that this advantage arises from MDMs' ability to simultaneously model all conditional distributions, making them less sensitive to distributional shifts compared to the unidirectional dependencies in ARMs. These results indicate that MDMs are inherently more robust to temporal shifts, making them better suited for evolving data distributions.

## 8 CONCLUSION

In this paper, we demonstrate the strong scalability of MDMs through a comprehensive scaling analysis. Our results show that MDMs can achieve comparable performance to ARMs in key tasks, such as language understanding, supported by the scaling law and the unsupervised classifier-free guidance. Furthermore, MDMs effectively address major limitations of ARMs, including the reverse curse and temporal quality degradation, even outperforming much larger models like Llama and GPT-3 in these aspects. These findings highlight MDMs as a promising alternative to ARMs for language modeling at scale.

We also observe that MDMs exhibit certain limitations, particularly in scaling laws and conditional generation, where a gap compared to ARMs persists. Our work provides a holistic view of the potential and limitations of MDMs, encouraging future research toward more efficient designs.

One of the most important future directions is to scale MDMs to larger sizes, potentially matching advanced ARMs (Achiam et al., 2023; Dubey et al., 2024). This would allow for a thorough investigation into the emergent behaviors (Wei et al., 2022a) and long-range reasoning capabilities (Wei et al., 2022b) of MDMs. By scaling up, we hope that MDMs can fully demonstrate their unique advantages over ARMs in real-world scenarios, offering a competitive alternative. Further, we believe the studies can deepen our understanding of large language models and the role of key factors such as autoregressive formulation in achieving such intelligence.

We also note another line of research focusing on continuous diffusion language models (Li et al., 2022; Gong et al., 2022; Han et al., 2022; Mahabadi et al., 2023; Strudel et al., 2022; Chen et al., 2022; Gulrajani & Hashimoto, 2024; Graves et al., 2023; Xue et al., 2024; Dieleman et al., 2022). However, the experiments in this domain are relatively small in scale and lack evaluation on standard language benchmarks. We hypothesize that MDMs enjoy better scalability than these models due to their alignment with the inherent structure of language and ARMs.

ETHICS STATEMENT

This paper focuses on the improvement of language models, which have vast potential to enhance communication, automate tasks, and facilitate access to information across languages. However, if misused, these models could be exploited to generate false information. Moreover, if trained on biased datasets, the generated text could perpetuate these biases. To mitigate these risks, we commit to transparency in our development processes and to continuously focus on research related to the safety and fairness of language models to further improve our models.

ACKNOWLEDGMENTS

This work was supported by the National Natural Science Foundation of China (No. 92470118); Beijing Natural Science Foundation (No. L247030); Beijing Nova Program (No. 20220484044); Major Innovation & Planning Interdisciplinary Platform for the "Double-First Class" Initiative, Renmin University of China; the Fundamental Research Funds for the Central Universities, and the Research Funds of Renmin University of China (22XNKJ13). The work was partially done at the Engineering Research Center of Next-Generation Intelligent Search and Recommendation, Ministry of Education.

We thank Jingyang Ou for the insightful discussions on RADD. We also thank Ang Lv for valuable conversations about the reverse curse and Wenkai Yang for discussions on the supervised fine-tuning of ARMs. Additionally, we appreciate Siqi Kou for providing guidance on data processing and evaluation for the conditional generation experiments.

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

---

**Algorithm 1** Greedy sampling method of MDMs

---

**Require:** A all masked sequence $x_1$ of length $L$, sampling steps $N$.
  1: **for** $t = 1, \frac{N-1}{N}, \frac{N-2}{N}, \ldots, \frac{1}{N}$ **do**
  2:      $s = t - \frac{1}{N}$
  3:      **for** $i = 0, 1, \ldots, L-1$ **do**
  4:         **if** $x_t^i \neq m$ **then**
  5:            $x_0^i = x_t^i, c^i = 1$
  6:         **else**
  7:            $x_0^i = \arg\max_j p_{\boldsymbol{\theta}}(x_0^i | \boldsymbol{x}_t)_j$, and denote $c^i = p_{\boldsymbol{\theta}}(x_0^i | \boldsymbol{x}_t)_{x_0^i}$.
  8:         **end if**
  9:      **end for**
 10:      $l = \lfloor L(1-s) \rfloor$                          # we set the number of unmasked tokens to $l$ in timestep $s$
 11:      **for** $i = 0, 1, \ldots, L-1$ **do**
 12:         **if** $c^i \in \text{top} - l\left(\{c^i\}_{i=0}^{L-1}\right)$ **then**
 13:            $x_s^i = x_0^i$
 14:         **end if**
 15:      **end for**
 16: **end for**
 17: **return** $x_0$

---

## A   GREEDY SAMPLING METHOD OF MDMS

We employ the sampling method of MaskGIT (Chang et al., 2022) as the greedy sampling strategy for MDMs. For completeness, we include the algorithm in Alg. 1 and provide the following intuitive explanation.

Let us first revisit the original sampling method for MDMs as described in Eq. (2). During each sampling step from time $t$ to $s$, if $x_t^i \neq m$ it remains unchanged. Otherwise, it retains the masked state with a probability of $\frac{s}{t}$, or transitions to $x_0^i \sim p_{\boldsymbol{\theta}}(x_0^i | \boldsymbol{x}_t)$ with a probability of $1 - \frac{s}{t}$. It is important to note that for all masked tokens $x_t^i$, they transition to corresponding $x_0^i$ with the same probability of $1 - \frac{s}{t}$.

Different from the original sampling method, MaskGIT (Chang et al., 2022) does not transition all masked tokens to their corresponding $x_0^i$ with the same probability of $1 - \frac{s}{t}$. Instead, it specifically selects masked tokens that exhibit the highest conditional probability $p_{\boldsymbol{\theta}}(x_0^i | \boldsymbol{x}_t)_{x_0^i}$ for transition to $x_0^i$.

## B   EXPERIMENTAL DETAILS

### B.1   REPRODUCIBILITY STATEMENT

We implement our experiments based on the TinyLlama (Zhang et al., 2024) codebase. We use the code provided by TinyLlama to preprocess the SlimPajama (Soboleva et al., 2023) dataset. Additionally, we use the code provided by CLLM (Kou et al., 2024) to preprocess the ShareGPT dataset. We employ the fictitious dataset provided by Berglund et al. (2023) and Fineweb dataset (Penedo et al., 2024) for the reverse curse and temporal quality degradation experiments, respectively. Besides, when test math reasoning ability, we use the augmented data (Deng et al., 2023) for training and GSM8K (Cobbe et al., 2021b) dataset for test. Because of their simplicity, we preprocess these four datasets by ourselves. We employ the lm-eval (Gao et al., 2024) and fast-chat (Zheng et al., 2023) framework to evaluate language understanding tasks and conditional generation, respectively. In Sec. 5, the pre-trained GPT-2 and TinyLlama models are provided by HuggingaFace. The corresponding links are detailed in Tab. 7.

Table 7: **Links for code and checkpoints.**

| | Link |
|---|---|
| GPT-2 model | https://huggingface.co/openai-community/gpt2-xl |
| TinyLlama model | https://huggingface.co/TinyLlama/tinyLLaMA-v1.1-checkpoints/tree/step-300000 |
| Llama2 model | https://huggingface.co/meta-llama/Llama-2-7b-hf |
| T5 model | https://huggingface.co/google-t5 |
| TinyLlama codebase | https://github.com/jzhang38/TinyLlama |
| CLLM codebase | https://github.com/hao-ai-lab/Consistency_LLM |
| SlimPajama dataset | https://huggingface.co/datasets/cerebras/SlimPajama-627B |
| Augmented GSM8K dataset | https://github.com/da03/implicit_chain_of_thought |
| GSM8K dataset | https://huggingface.co/datasets/openai/gsm8k |
| ShareGPT dataset | https://huggingface.co/datasets/anon8231489123/ShareGPT_Vicuna_unfiltered |
| Reverse curse dataset | https://huggingface.co/datasets/lberglund/reversal_curse |
| Fineweb dataset | https://huggingface.co/datasets/HuggingFaceFW/fineweb |
| Lm-eval framwork | https://github.com/EleutherAI/lm-evaluation-harness |
| Fast-chat framework | https://github.com/lm-sys/FastChat |

## B.2 ADDITIONAL EXPERIMENTAL DETAILS OF ISOFLOP ANALYSIS

**Training details.** We use identical optimizer settings for both MDMs and ARMs during pre-training. Consistency with TinyLLama (Zhang et al., 2024), we utilize the AdamW optimizer (Loshchilov, 2017), setting $\beta_1 = 0.9$, $\beta_2 = 0.95$, and a weight decay of 0.1. Additionally, we apply a cosine learning rate schedule with a maximum learning rate of $4 \times 10^{-4}$ and a minimum learning rate of $4 \times 10^{-5}$ with $1\%$ of the tokens for linear warmup. Notably, if the number of warmup steps is less than 100, it is set to 100. The batch size is set to 256.

Apart from the scaling law experiment, we also pre-train two 1.1B MDMs using $1.6 \times 10^{21}$ and $3.3 \times 10^{21}$ FLOPs for downstream tasks. Except for the batch size, we adopt the aforementioned pre-training settings for the 1.1B model. Specifically, we set the batch size to 384 and 1024 for the models trained with $1.6 \times 10^{21}$ and $3.3 \times 10^{21}$ FLOPs, respectively, due to the differing number of GPUs utilized. In the first version of this paper, we used the MDM with $1.6 \times 10^{21}$ pre-training FLOPs in Sec. 5-6 and Sec 7.1. In the second version (this version), we replaced the MDM in Sec. 5 with the MDM pre-trained with $3.3 \times 10^{21}$ FLOPs to more comprehensively demonstrate the scaling performance of the MDM.

**Evaluation details.** For MDMs, we found that using more Monte Carlo estimation samples (i.e., 128) when computing the validation loss effectively reduces the number of outliers in Fig. 1b. This is because increasing the number of Monte Carlo samples reduces the variance of the estimation, leading to a more precise estimation of the validation loss.

**Model configs.** We list all model configurations in Tab. 8.

## B.3 ADDITIONAL EXPERIMENT DETAILS OF LANGUAGE UNDERSTANDING

We use the 1.1B MDM with $3.3 \times 10^{21}$ pre-training FLOPs (see details in Appendex B.2) in Sec. 5. This model is pre-trained with $1\%$ data set to random length.

Additionally, we provide the experimental details for the GSM8K results. We fine-tune the MDM on the augmented training data (Deng et al., 2023) for 40 epochs, following prior works (Ye et al., 2024; Gong et al., 2024). The optimizer settings remain consistent with those described in Appendix B.2, and each data instance is padded with |EOS| to a length of 256 tokens. For evaluation, we use greedy sampling, setting the sampling steps to 256 and applying an unsupervised CFG scale of 0.1.

## B.4 ADDITIONAL EXPERIMENTAL DETAILS OF CONDITIONAL GENERATION

**Setup.** We use identical optimizer settings for both MDMs and ARMs during supervised fine-tuning. Similar to our pretraining process, we use the AdamW optimizer (Loshchilov, 2017) with hyperparameters $\beta_1 = 0.9$, $\beta_2 = 0.95$, and a weight decay of 0.1. We employ a cosine learning rate schedule starting from a maximum learning rate of $2 \times 10^{-4}$ and decaying to a minimum of

Table 8: **Model configurations of MDMs and ARMs.** [*] labels the non-embedding parameters.

| Parameters[*] (M) | n_layers | n_heads | n_embed | intermediate_size |
|---|---|---|---|---|
| 19 | 8 | 6 | 384 | 1536 |
| 34 | 8 | 8 | 512 | 2048 |
| 48 | 9 | 9 | 576 | 2304 |
| 66 | 10 | 10 | 640 | 2560 |
| 75 | 16 | 8 | 640 | 1600 |
| 85 | 13 | 10 | 640 | 2560 |
| 113 | 12 | 12 | 768 | 3072 |
| 142 | 15 | 12 | 768 | 3072 |
| 170 | 18 | 12 | 768 | 3072 |
| 180 | 14 | 14 | 896 | 3584 |
| 206 | 16 | 14 | 896 | 3584 |
| 231 | 18 | 14 | 896 | 3584 |
| 268 | 16 | 16 | 1024 | 4096 |
| 302 | 18 | 16 | 1024 | 4096 |
| 336 | 20 | 16 | 1024 | 4096 |
| 472 | 18 | 10 | 1280 | 5120 |
| 551 | 21 | 10 | 1280 | 5120 |
| 571 | 18 | 11 | 1408 | 5632 |
| 629 | 24 | 10 | 1280 | 5120 |
| 666 | 21 | 11 | 1408 | 5632 |
| 717 | 19 | 12 | 1536 | 6144 |
| 761 | 24 | 11 | 1408 | 5632 |
| 831 | 22 | 12 | 1536 | 6144 |
| 944 | 25 | 12 | 1536 | 6144 |
| 1028 | 20 | 14 | 1792 | 7168 |
| 1233 | 24 | 14 | 1792 | 7168 |
| 1476 | 22 | 16 | 2048 | 8192 |
| 1678 | 25 | 16 | 2048 | 8192 |
| 2121 | 28 | 17 | 2176 | 8704 |

Table 9: **Overview of different CFG strategies for conditional generation.** The standard CFG fine-tunes both conditional and unconditional distributions on paired data, while unsupervised CFG is enhanced by fine-tuning only conditional distribution. Unsupervised CFG already leverages large-scale pre-trained data to obtain a strong unconditional model, resulting in improved performance compared to standard CFG.

| | Training | | Sampling |
|---|---|---|---|
| | Conditional | Unconditional | |
| No-CFG | ✓ | ✗ | w/o CFG |
| Standard CFG | ✓ | ✓ | w/ CFG (i.e., Eq. (7)) |
| Unsupervised CFG | ✓ | ✗ | w/ CFG (i.e., Eq. (7)) |

$2 \times 10^{-5}$. Additionally, we apply linear warm-up over the first 200 steps and set the batch size to 256.

For the preprocessing of the ShareGPT dataset, we use the same method as described in Kou et al. (2024). In addition, in line with Kou et al. (2024), we fine-tune both ARMs and MDMs on the first-turn conversation from the ShareGPT dataset and report the first-turn conversation score. We do not use any annealing sampling method for ARMs and MDMs during generation. The MT-Bench score is obtained via the "gpt-4o-2024-05-13" API provided by OpenAI.

**Different CFG strategies.** We provide an overview of no CFG, standard CFG, and unsupervised CFG in Tab. 9.

Table 10: **Comparison of different methods to address train-test discrepancy.** 1% and 5% denote that set 1% and 5% training data to random length, respectively. For simplicity, we employ the chain rule to calculate the conditional likelihood and do not use the unsupervised CFG. Both variable length training and padding mask tokens significantly improve the performance of MDMs in language understanding tasks.

| | ARC-Easy | BoolQ | Hellaswag | OpenBookQA | PIQA | RACE | SIQA | LAMBADA |
|---|---|---|---|---|---|---|---|---|
| Original | 30.13 | 55.29 | 29.16 | 26.20 | 56.04 | 28.52 | 35.21 | 16.51 |
| Padding | **38.38** | 59.91 | 31.63 | **27.60** | **60.77** | 28.42 | **37.00** | 31.03 |
| 1% | 37.79 | **61.50** | 31.86 | 27.00 | 60.34 | **29.19** | 36.85 | **36.00** |
| 5% | 37.12 | 51.87 | **32.29** | 26.60 | 58.98 | 29.18 | 36.85 | 32.04 |

During fine-tuning on labeled data, the standard CFG (Ho & Salimans, 2022) replaces the label with a special token with a probability of 10%. This special token represents the unconditional distribution, thereby enabling the simultaneous training of both conditional and unconditional distributions. Specifically, for the implementation of standard CFG in our experiment, we randomly replace the prompt with the masked tokens with probability 10%.

In contrast to the standard CFG, unsupervised CFG already leverages large-scale pre-trained data to obtain a strong unconditional model, therefore we only enhance its conditional distribution during fine-tuning on paired data.

During inference, both standard CFG and unsupervised CFG employ the rescaled conditional distribution defined in Eq. (7).

### B.5 ADDITIONAL EXPERIMENTAL DETAILS OF REVERSE CURSE

We use the same optimizer settings as Appendix B.4 except batch size when finetuning on the fictitious dataset provided by Berglund et al. (2023). As the fictitious dataset is smaller (i.e., only 3600 data), we use a batch size of 32 for fine-tuning. We train for 10 epochs following Berglund et al. (2023). We also pad each sample with the |EOS| token to the maximum sequence length within a batch as detailed in Sec. 6. Following the same approach as Berglund et al. (2023), we do not mask the loss on prompts, adding noise to the prompt and response simultaneously as Eq. (4).

For the T5 model, we adopted the same settings as MDM, except for the learning rate. Initially, we tested maximum learning rates in $\{10^{-5}, 10^{-4}, 10^{-3}, 10^{-2}\}$ and found that $10^{-4}$ yielded the best results. We further refined the learning rate by experimenting with $\{2 \times 10^{-5}, 3 \times 10^{-5}, 5 \times 10^{-5}, 2 \times 10^{-4}, 3 \times 10^{-4}, 5 \times 10^{-4}\}$, identifying $2 \times 10^{-4}$ as the optimal maximum learning rate. The minimum learning rate was set to one-tenth of the maximum.

## C ADDITIONAL RESULTS

### C.1 MORE RELATED WORK ABOUT UNSUPERVISED CFG.

Prior studies Zhao et al. (2021); Holtzman et al. (2021) emphasize the importance of unconditional distributions in large language models. By exploiting the distinctive properties of MDMs, unsupervised CFG introduces a novel approach to estimating the unconditional distribution (i.e., Eq. (8)) and incorporates this estimate through an alternative mechanism (i.e., Eq. (7)), inspired by the standard CFG framework (i.e., Eq. (6)).

### C.2 ADDITIONAL RESULTS OF LANGUAGE UNDERSTANDING

**Results of fixing traing-test discrepancy.** For efficiency, we employ MDM with 220M parameters, pre-trained for $10^{20}$ FLOPs to experiment. Tab. 10 presents the ablation studies of variable length training and padding mask tokens, demonstrating that both methods significantly improve the performance of MDMs.

Table 11: **Comparison of different likelihood evaluation methods.** We employed 1024 Monte Carlo samples for the Monte Carlo estimation. All results are reported with the corresponding optimal unsupervised CFG scale. The optimal likelihood evaluation method differs across tasks.

|  | ARC-Easy | BoolQ | Hellaswag | OpenBookQA | PIQA | RACE | SIQA |
|---|---|---|---|---|---|---|---|
| Monte Carlo | **39.02** | **62.17** | **34.10** | 30.40 | 59.14 | **30.81** | **37.41** |
| Chain rule | 37.88 | **62.17** | 32.20 | **34.20** | **60.39** | 29.67 | 37.10 |

Table 12: **MT-Bench results of MDM with $10^{20}$ pre-training FLOPs.**

|  | CFG = 0.4 | CFG = 0.6 | CFG = 0.8 |
|---|---|---|---|
| Score | 1.21 | 1.22 | 1.23 |

Table 13: **Additional ablation results on unsupervised CFG.** The sampling computation in each column is the same.

|  | Mt-Bench score (sampling steps) | |
|---|---|---|
| w/o CFG | 1.32 (256) | 1.35 (512) |
| w/ CFG | 1.56 (128) | 1.60 (256) |

**Results of different likelihood evaluation methods.** For efficiency, we employ MDM with 220M parameters, pre-trained for $10^{20}$ FLOPs, and set 1% training data to random length. Tab. 11 presents the ablation studies of different likelihood evaluation methods.

We empirically find that tasks requiring step-by-step reasoning tend to achieve higher accuracy when using the chain rule for likelihood evaluation. In contrast, tasks focused on contextual understanding perform better with Monte Carlo estimation. A comprehensive study is left for future work.

**Scaling behavior of MDMs on language understanding tasks.** As shown in Fig. 3, the performance of MDMs on the language understanding tasks shows a scaling behavior with respect to the validation loss, which is consistent with observations in ARMs (Du et al., 2024). For efficiency and simplicity, methods for fixing train-test discrepancies and unsupervised CFG are not applied in this analysis.

### C.3 Additional Results of Conditional Generation

**More MT-Bench results of MDM.** In Sec. 6, we report the MT-Bench results of ARM and MDM with $10^{20}$ and $1.6 \times 10^{21}$ pre-training FLOPs, respectively. Here, we present the MT-Bench result of MDM with $10^{20}$ pre-training FLOPs in Tab. 12.

**Additional ablation results on unsupervised CFG.** Table 13 shows that unsupervised CFG outperforms sampling without CFG with half the sampling steps (i.e., equal sampling computation).

**Generated sentence of MDM on MT-Bench.** We present some answers generated from MDM in Fig. (4-6).

### C.4 Additional Results of Reverse Curse

**Complementary explanation.** Kitouni et al. (2024) highlight that models trained with less dependence on the precise sequence of tokens can successfully mitigate the reverse curse, which serves as a complementary explanation for the findings of Table 5.

In the NameToDescription test data for the same direction of Table 5, the T5 model outperforms MDM in BLEU scores but lags in exact match accuracy. This is because the T5 model tends to produce responses that are similar to the ground truth but differ slightly in a few words. It is worth

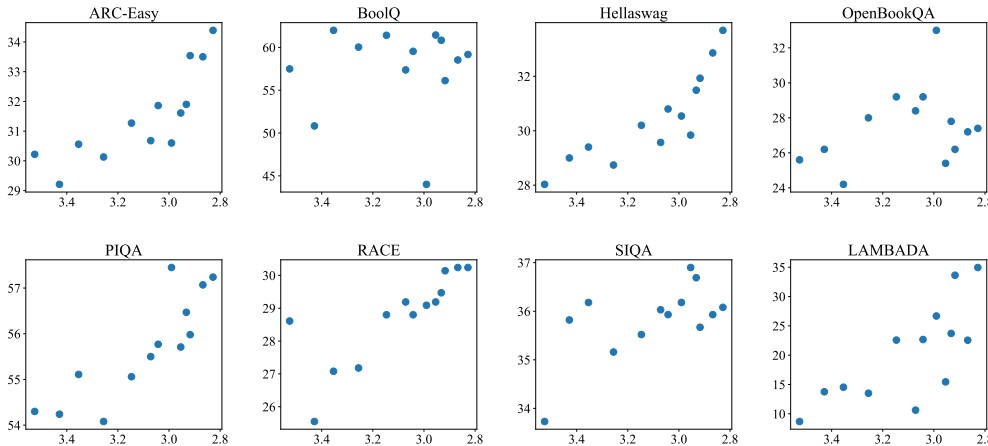

Figure 3: **Scaling properties of MDMs on language understanding tasks.** The x-axis represents the validation loss, while the y-axis indicates the accuracy.

Table 14: **Effectiveness of unsupervised CFG on reverse curse.** The unsupervised CFG enhances the performance of MDM on the reverse queries.

| | DescriptionToName | | NameToDescription | | | |
|---|---|---|---|---|---|---|
| | Same direction | Reverse direction | Same direction | | Reverse direction | |
| | Acc. ↑ | Acc. ↑ | Acc. ↑ | BLEU ↑ | Acc. ↑ | BLEU ↑ |
| w/o CFG | 95 | 85 | **52** | **80** | 28 | 60 |
| w/ CFG | **97** | **92** | 49 | 76 | **37** | **67** |

noting that reverse question data shows a larger divergence between the training and testing sets, which accounts for the performance decline of MDM in the reverse task.

**Additional results.** Tab 14 shows the effectiveness of the unsupervised CFG on the reverse curse.

## D    EVALUATION METRICS

In this section, we provide an overview of the benchmarks used in Sec. 5 and show some cases from these benchmarks in Tab. 15.

**ARC-Easy.** A subset of the **AI**2 **R**easoning **C**hallenge that focuses on elementary-level science questions to evaluate the model's reasoning ability through basic scientific concepts.

**BoolQ.** A yes-or-no question-answering dataset designed to evaluate the model's ability to answer questions based on a given passage.

**HellaSwag.** A metric assesses the model's commonsense reasoning ability by completing a given sentence with one of four options.

**OpenBookQA.** A question-answering dataset, modeled after open-book exams, is designed to assess a model's understanding of a subject by requiring multi-step reasoning and the integration of additional commonsense knowledge.

**PIQA. P**hysical **I**nteraction **Q**uestion **A**nswering is a metric that evaluates physical reasoning ability by asking models to select the best solution to a given problem involving everyday physical scenarios.

Table 15: **Examples from language understanding benchmarks.**

| Metric | Question | Choices or answers |
|---|---|---|
| ARC-Easy | Which of the following was probably most important in the formation of dark, fertile soil that is good for farming? | A. plant decomposition
B. radioactive decay
C. water erosion
D. wind erosion |
| BoolQ | was the leaning tower of pisa built leaning | Yes
No |
| HellaSwag | A camera pans around a set of stairs and leads into people working out in a class. Several shots are shown of people working out together while a man speaks to the camera. the man | A. continues speaking while more people are shown working out together.
B. is seen crashing into a wall several more times while people watch on the side.
C. then leads the group on a liquid workout together.
D. continues speaking to the camera while more shots are shown of them lifting weights and/or speaking to the camera. |
| OpenBookQA | A man plugs his television into an outlet behind a cabinet. He sees that the television may now be turned on so that he can watch his favorite show. The man knows that by hooking the t.v. cord into the outlet | A. he completed a lap
B. he made a good deal
C. he invented new circuits
D. he completed a circuit |
| PIQA | When boiling butter, when it's ready, you can | A. Pour it onto a plate
B. Pour it into a jar |
| SIQA | Taylor took the poor dog she found on the road to the vet. What will the vet want to do next? | A. pronounce the dog dead
B. wraps the dog in a towel
C. check to make sure the dogs okay |
| RACE | The article is too long and has been disregarded. According to the passage, which of the following statements is TRUE? | A. There is more petroleum than we can use now.
B. Trees are needed for some other things besides making gas.
C. We got electricity from ocean tides in the old days.
D. Gas wasn't used to run cars in the Second World War. |
| LAMBADA | "Again, he left that up to you. However, he was adamant in his desire that it remain a private ceremony. He asked me to make sure, for instance, that no information be given to the newspaper regarding his death, not even an obituary. I got the sense that he didn't want anyone, aside from the three of us, to know that he'd even _. | died |
| GSM8K | Weng earns $12 an hour for babysitting. Yesterday, she just did 50 minutes of babysitting. How much did she earn? | 10 |

**SIQA.** **S**ocial **I**nteraction **Q**uestion **A**nswering is a benchmark for commonsense reasoning and is established by presenting scenarios that require reasoning about social interactions and the motivations behind human behavior.

**RACE.** **R**e**A**ding **C**omprehension Dataset From **E**xaminations was designed to evaluate reading comprehension ability by understanding and interpreting text at a high school level.

**LAMBADA.** A dataset to evaluate models' capabilities in text understanding through a final single-word prediction task based on a given context.

**GSM8K.** GSM8K (Grade School Math 8K) is a high-quality dataset of grade school math word problems, created to enable question answering involving multi-step mathematical reasoning.

**Prompt:**
A chat between a curious user and an artificial intelligence assistant. The assistant gives helpful, detailed, and polite answers to the user's questions. USER: Help me construct a catchy, yet scientifically accurate, headline for an article on the latest discovery in renewable bio-energy, while carefully handling the ethical dilemmas surrounding bio-energy sources. Propose 4 options. ASSISTANT:

**Answers:**
*Option 1: " zum Uncovering the Power of Renewable Bio-energy: The Future We Have and Our Right to Utilize"
This article should handle the ethical dilemmas surrounding bio-energy sources, including addressing the need for sustainable alternatives, the choices made by developing renewable bio-energy sources, and the necessary balance the right of honest and technology.
*Option 2: "Exploring the latest Sustainable Energy for All: Majestic Renewable Bio-Earth Reports and Stories"/"Comprehensive reports and insights on the latest research in renewable bio-energy sources. This may have a focus on renewable energy but should also be sustainable.
*Option 3: "Unleashing Record Renewable Energy Production: Exploring Beyond Coal Stream Options" Record renewable energy production capacity and conditions for development. This article focuses on the future of renewable energy sources, while addressing the challenges involved with sustainatory alternatives.
*Option 4: "wissenschaft Handling the Environmental Impact of Renewable Energy: Harnessing the opportun for Renewable Energy Sources"
One of the ethical dilemmas surrounding bio-energy sources are economic and social dilemmas, such as neglecting local, regional, or communities. This article seeks to discuss how to develop renewable energy sources that reduce the environmental impact and promote sustainability.
*Option 5: ": Exploring the Future of Renewable Bio-energy: Navigating the Environmental Impacts and Source Options"
Discussing decision-making about the practices and consequences of developing renewable energy sources while ensuring a balance between ethical dilemmas and sustainability.
*Option 6: "Exploring the Renewable Bio-Earth: Unlocking the Future of Renewable Energy Sources"

Figure 4: Generated sentence of MDM on the MT-Bench (case 1).

**Prompt:**
A chat between a curious user and an artificial intelligence assistant. The assistant gives helpful, detailed, and polite answers to the user's questions. USER: What are some business etiquette norms when doing business in Japan? ASSISTANT:

**Answers:**
Business etiquette is influenced by Western culture, but Japan has many norms of respect when interacting. Here are some etiquette norms when doing business in Japan:
1. Show respect: Always greet with greets in a respectful manner in a Japanese business: Always greet with respectful and professional lines when working with Japanese service workers. Always greet the customers politely unless they have a reason.
3. Practice etiquette: Always make respectful statement and be polite: Always greet someone in a respectful manner as it is the cultural norm and always respond respectfully and in a polite manner.
4. Always work courteously: Always respect when hideholding in Japanese business: Always respect when dealing with Japanese service workers or officials. Always show respect when asking for something unless it is courteous and building.
5. Use Japanese apparel and respect Japanese culture when doing business in a Japanese company: Do not use Japanese or Western companies or meet Japanese officials unless you are allowed to use their language and respect in Japanese business. While it is common to use monosma garments, it is not in Japan keeping dignity: Always respect and maintain respect: Always respect and maintain your dignity when dealing with Japanese service workers or especially when answering questions or making demonstrations of professional respect or respect
7. Request permission: Alwaysfshow respect when asking permission: Request permission respectfully, fairly and courteously to avoid a request. Always show respect when asking for permission unless it is haing.
9. Show respect: Always respect the customer with respect in Japan business: Always respect the customer with respect. These are a few examples of common business etiquette norms of respect when doing business in Japan. In general, business etiquette norms depend on the specific industry or branch of operation and may vary depending on your understanding and circumstances.

Figure 5: Generated sentence of MDM on the MT-Bench (case 2).

**Prompt:**
A chat between a curious user and an artificial intelligence assistant. The assistant gives helpful, detailed, and polite answers to the user's questions. USER: Compose an engaging travel blog post about a recent trip to Hawaii, highlighting cultural experiences and must-see attractions. ASSISTANT:

**Answers:**
An enchanted-eyed traveler recently had the pleasure of discover Hawaii, a tropical paradise in the heart of the Pacific. The trip offers a diverse and blend cultural experience, representing the culture and outdoor adventure that Hawaii calls home.
The first thrilling experience is I hadn't seen a wave breaking before in these green waters. But I experienced the same thing and witnessed the cool water rushing past me and the surf noticing caught my attention while peeking at the tall towering waterfalls.
Another cultural experience I had in my trip was exploring the land and its beaches through snorkeling. It is not just snorkeling, it is discovering beach dunes and sunnaporkeling on the gorgeous sand of Hana Beach, which is more famous than its name. Whether you can surf good, hike, or monke, it was a day of exploration.
My trip to Hawaii was a diverse destination with many aspects to explore and not enough time to see everything. Each culture has its own unique culture, determining the must see attractions. If you can only take one trip, come here and explore the beauty.

Figure 6: Generated sentence of MDM on the MT-Bench (case 3).

