# OpenReview forum: "Scaling up Masked Diffusion Models on Text"
_ICLR.cc/2025/Conference — ICLR 2025 Poster_

### Official Review · Reviewer_6Chg · 2024-10-16

**Soundness:** 3
**Presentation:** 3
**Contribution:** 3
**Rating:** 6
**Confidence:** 4

**Summary:**

This paper explores the potential of Masked Diffusion Models (MDMs) in language tasks, highlighting their scalability and performance compared to autoregressive models (ARMs). The authors train models with up to 1.1 billion parameters and demonstrate a scaling law for MDMs. MDMs show competitive results in language understanding and provide a flexible trade-off in conditional generation compared to ARMs. Additionally, the authors find MDMs are robust to temporal shifts in data and overcome the reverse curse problem faced by ARMs.

**Strengths:**

- Scaling MDM: The main contribution of this paper lies in scaling mask discrete diffusion (MDM) and provide the corresponding scaling law.
- Identifying distinctive features of diffusion: The authors highlight how diffusion can effectively address shortcomings like temporal shifts in data and the reverse curse found in ARM. This paves the way for additional research and application opportunities in the realm of diffusion language models.

**Weaknesses:**

- The paper primarily compares MDM and AR, but lacks a clear comparison with other pretrained diffusion language models such as Plaid (https://arxiv.org/abs/2305.18619) and SEDD (https://arxiv.org/abs/2310.16834) under the same setting.
- The diffusion loss provides an upper bound and should not be directly compared with the loss in AR models. The comparison in Figure 2(a) appear to lack rigor, a similar problem for the perplexity comparison detailed in Table 7.
- About the setup described in Table 6, it appears that MDM fine-tuned on the dataset, while GPT3 and LLaMA not. To mitigate the potential confounding variable of fine-tuning versus prompting, a baseline model like fine-tuned GPT-2 ought to be included for a more comprehensive analysis.
- The proposed unsupervised CFG is similar to calibration in AR (http://arxiv.org/abs/2102.09690,http://arxiv.org/abs/2104.08315), but the relationship is not properly discussed.

Minor:
- typo: line 756 “Greddy sampling“

**Questions:**

- Why different likelihood evaluation methods result in distinct behavior on different tasks? How can we decide on the fly given a new task?
- What's the relationship between 10^20 Flops and tokens? Do both ARM and MDM consume the same amount of tokens?
- What’s the inference timesteps in Table 5? Line 409 “Built upon the unsupervised CFG”, does it means the inference timesteps are doubled? It's better to draw a curve to show the speed advantage of diffusion, which I guess is in generating longer texts.
- What’s the computation resource and GPU hours comparison for MDM and AR?

---

> ### Author Response · Authors · 2024-11-21
> **Response to Reviewer 6Chg [1/2]**
>
> We thank reviewer 6Chg for the acknowledgment of our contributions and the valuable comments. We respond to your questions and concerns below.
>
> ## Weakness 1: Comparison with other diffusion language model
>
> Thank you for your valuable comment. We would like to clarify that **our goal is not to propose a new algorithm that outperforms existing diffusion language models.** Instead, we focus on exploring the scalability of the best existing diffusion language models (i.e., RADD [1], as discussed in Lines 134-137) and comparing them with ARMs. We believe that the contributions of our paper, such as the scaling laws, are orthogonal to algorithm design and can help scale up other models. Whether scaled Plaid or SEDD outperforms RADD or not does not alter the significance of our contribution.
>
> As for a clear comparison in the same setting, existing results (Tab 1 in [1]) have proven that RADD outperforms Plaid and SEDD in terms of zero-shot perplexity.
>
>
> [1] Ou et al. Your Absorbing Discrete Diffusion Secretly Models the Conditional Distributions of Clean Data. arXiv 2024.
>
> ## Weakness 2: The loss of MDMs is an upper-bound loss and should not be directly compared with the loss in ARMs.
>
> Thank you for your valuable comment. Theoretically, the upper bound of the perplexity is a meaningful guarantee of the perplexity (or negative log-likelihood) because the objective is to minimize the perplexity. Therefore, in the literature of both the image and text domains [2, 3, 4, 5, 6], **the upper bound of the negative likelihood for diffusion models is widely adopted to compare with the exact negative likelihood of autoregressive models.**
>
> Besides, as suggested by previous works [3], **the upper bound is typically tight (i.e. very close to the negative log-likelihood) in practice.**
>
> [2] Ho et al. Denoising Diffusion Probabilistic Models. NeurIPS 2020.
>
> [3] Kingma et al. Variational Diffusion Models. NeurIPS 2021.
>
> [4] Austin et al. Structured Denoising Diffusion Models in Discrete State-Spaces. NeurIPS 2021.
>
> [5] Gulrajani et al. Likelihood-Based Diffusion Language Models. NeurIPS 2023.
>
> [6] Lou et al. Discrete Diffusion Modeling by Estimating the Ratios of the Data Distribution. ICML 2024.
>
> ## Weakness 3: Experiment setup in Table 6.
>
> Thank you for your valuable question. As presented by previous work [8, 9], both GPT3 and Llama2 are fine-tuned on the fictitious dataset for 10 epochs. Following [8, 9], we also fine-tune MDM on **the same dataset** for 10 epochs. We will detail the settings in the revision.
>
> ## Weakness 4: More related works
>
> Thank you for pointing out these additional related works. We will include a discussion of both in the revision.
>
> Work [7] uses content-free input to calibrate the bias of LLMs through an affine transformation, while [8] proposes the Domain Conditional Pointwise Mutual Information (DCPMI) method, which improves LLMs by using a domain premise string to estimate the unconditional probability.
>
> Both of these previous works [7, 8] and our unsupervised CFG highlight the importance of unconditional models in language tasks. However, **our unsupervised CFG differs in that it leverages the unique characteristics of MDMs to propose a new method for estimating the unconditional model.** Additionally, **we introduce a different approach to incorporating the unconditional model**, inspired by the CFG used in image diffusion models [9].
>
> [7] Zhao et al. Calibrate Before Use: Improving Few-Shot Performance of Language Models. ICML 2021.
>
> [8] Holtzman et al. Surface Form Competition: Why the Highest Probability Answer Isn't Always Right. EMNLP 2021.
>
> [9] Ho et al.  Classifier-free diffusion guidance. NeurIPS 2021.

---

> ### Author Response · Authors · 2024-11-21
> **Response to Reviewer 6Chg [2/2]**
>
> ## Weakness 5: Minor typo
> Thank you for carefully reading. We will fix this typo in the revision.
>
>
> ## Question 1: Why do different likelihood evaluation methods result in distinct behavior on various tasks?
>
> Thank you for your insightful question. We have some empirical observations and will conduct more in-depth research in future work.
>
> We found that for tasks requiring step-by-step reasoning, using the chain rule for likelihood evaluation may result in higher accuracy. For example, in the PIQA benchmark, one question asks: *'To separate egg whites from the yolk using a water bottle, you should'* with the corresponding answer: *'Squeeze the water bottle and press it against the yolk. Release, which creates suction and lifts the yolk.'* This task requires the model to understand the physical world and perform step-by-step reasoning, which may explain why using the chain rule leads to higher accuracy on PIQA.
>
> However, for tasks that focus on contextual understanding, using Monte Carlo estimation may yield better accuracy. For instance, in the ARC-Easy benchmark, one question asks: *'Which of the following was probably most important in the formation of dark, fertile soil that is good for farming?'* with the corresponding answer: *'plant decomposition.’* This question emphasizes the model's common-sense knowledge and contextual understanding, which may explain why Monte Carlo estimation results in better accuracy on ARC-Easy.
>
> We believe that these observations offer valuable guidance for exploring new datasets. Additionally, one could sample a small portion of the data as a validation set and use it to determine the most effective likelihood evaluation method. A comprehensive, systematic study is left for future work.
>
> ## Question 2: Relationship between training Flops and tokens?
>
> Thank you for the valuable question. The relationship between training Flops and tokens **is the same** in ARMs and MDMs. In particular, as explained in Lines 184-185, we have the relation $C=6ND$ for both models, where $C$ denotes the training FLOPs, $N$ is the non-embedding parameters, and $D$ is the number of tokens.
>
> For instance, in Table 2, both the 220M ARM and MDM (with 170M non-embedding parameters) are trained for $10^{20}$ FLOPs, which means both models are trained on 100B tokens.
>
> We will make it clearer in the revision.
>
> ## Question 3: Inference timesteps
>
> Thank you for your valuable question. When using unsupervised CFG, the conditional and unconditional models are computed in parallel within a single forward pass. Consequently, the computational cost of sampling doubles when employing CFG, given the same number of sampling steps.
>
> As shown in the table below, we can see that unsupervised CFG outperforms the baseline with the same sampling computation, i.e., unsupervised CFG uses only half the sampling steps. The data in the table is presented in the format: `MT-Bench Score (Sampling Steps)`.
>
> | w/o CFG | 1.32 (256) | 1.35 (512） |
> | --- | --- | --- |
> | w/ CFG | 1.56 (128） | 1.60 (256） |
>
> Besides, thank you for suggesting a curve to illustrate the speed of generating longer texts. However, once MDMs and ARMs are trained, the length of the generated sentences is determined by the underlying data distribution. Exploring the generation of varying sentence lengths would require training on datasets with different sequence lengths. Due to time constraints, we leave this exploration for future work.
>
> ## Question 4: GPU hour comparison
>
> Thank you for your valuable question. **Given the same model size and training data, the training costs for ARM and MDM are identical**. The quantitative relationship can be found in the response to Question 2.
>
> This is because both ARM and MDM use a Transformer architecture, and the only difference during a forward pass is that ARM requires adding a causal mask to the attention map, while MDM does not. However, the computational cost of this operation is negligible.
>
> We will make it clearer in the revision.

---

> > ### Author Response · Authors · 2024-11-27
> >
> > Dear Reviewer 6Chg,
> >
> > Thank you once again for your valuable comments. We believe that our responses have addressed your concerns comprehensively. Could you kindly share whether our replies meet your expectations?
> >
> > Best regards,
> >
> > The Authors

---

> > > ### Comment · Reviewer_6Chg · 2024-11-27
> > >
> > > Thanks for the comprehensive responses. After reading the responses and the other reviews, I have decided to maintain my original score.

---

> ### Author Response · Authors · 2024-11-27
>
> We are happy that your concerns have been addressed. Thank you for your time and acknowledgment of our contributions.

---

### Official Review · Reviewer_3NzV · 2024-11-03

**Soundness:** 2
**Presentation:** 2
**Contribution:** 3
**Rating:** 6
**Confidence:** 3

**Summary:**

The paper focuses on scaling Masked Diffusion Models (MDMs) as an alternative to Autoregressive Models (ARMs) in language modeling.  The study establishes the first scaling law for MDMs, indicating that they can achieve performance on par with ARMs at a smaller computational cost gap than previously assumed. Moreover, MDMs demonstrate competitive performance in zero-shot language understanding tasks and effectively address ARM challenges like "reversal curse" and temporal quality degradation.

**Strengths:**

-   The introduction of scaling law for MDMs trained from scratch suggests that they show the same scaling potential as ARMs.
-   The presentation is clear and well-organized, with only minor typos.
-   Results on reversal curse tasks of MDMs is strikingly better than ARMs. This is an interesting point.

**Weaknesses:**

\textbf{Unfair Comparison:} Line 368-369: The authors extend MDM pre-training time by a factor of 16 to achieve a “meaningful” comparison with ARMs. However, a more valid comparison would involve MDMs and ARMs trained with equivalent compute budgets or trained to convergence for a given model size. Comparing an MDM to an ARM trained with only 1/16 of its FLOPs does not yield a balanced assessment.

\textbf{Missing Citations:} The paper lacks a related work section, making it difficult for readers unfamiliar with text DMs. Notably, a closely related work [1] is omitted. This prior research examines scaling trends of MDMs fine-tuned from a pretrained MLM and evaluates MDMs on complex tasks, aligning closely with this paper's contributions.

\textbf{Weak Benchmarks:} The benchmarks used in section 5 are mostly language understanding tasks, i.e, can be mostly formulated as a one-token-completion task. These NLU tasks are of less interest now, compared to summary benchmarks like gigaword and reasoning benchmarks like GSM8K, which require more complex abilities.

\textbf{Typos:}

Line 458 RELEVING should be relieving

Line 756 & 777 greddy should be greddy



[1] Diffusion Language Models Can Perform Many Tasks with Scaling and Instruction-Finetuning https://arxiv.org/pdf/2308.12219

**Questions:**

It is acceptable that MDMs trained from scratch do not match the performance of ARMs, as suggested by the scaling law in this paper as well as previous text DM work. I think a better way to present these findings should be reporting both the pros and cons of MDMs, e.g., they are more computation-consuming but can perform some tasks that state-of-the-art ARMs cannot do at all.

---

> ### Author Response · Authors · 2024-11-21
> **Response to Reviewer 3NzV [1/2]**
>
> We thank reviewer 3NzV for acknowledging our contributions and insightful comments. Below, we provided a point-to-point response to all comments.
>
> ## Weakness 1: MDM employs 16 times more training time than ARM in the conditional generation experiment
>
> Thank you for your valuable comment. We would like to kindly clarify that we did not claim MDMs outperform ARMs in conditional generation under same compute budget. As stated in Lines 369–370, MDMs utilized 16 times the pre-training time in Table 5, and we have provided the results with the same compute budget in the submission, as mentioned in Lines 369-370 and detailed in Appendix C.2 (Lines 944–946).
>
> In contrast, **we aim to explore the potential of MDMs in tackling challenging open-ended questions.** The results in Table 5 suggest that MDMs offer a flexible trade-off in conditional generation at the cost of 16 times the pre-training time, without any systematic optimization. This result highlights the potential of MDMs and may inspire further research on more computationally efficient MDMs, as acknowledged by Reviewer cmg3.
>
> We appreciate your concern about misunderstanding for readers. **To address this issue, we will highlight the computation of MDMs in Table 5 and add the above discussion in the conclusion section of the revision.**
>
> ## Weakness 2: More related work
>
> Thank you for pointing out the related work [1]. We acknowledge that it examines scaling trends of MDMs fine-tuned from a pre-trained XLM-RoBERTa [2, 3]. In comparison, our paper has distinct contributions.
>
> 1. We focus on training an MDM from scratch while the related work [1] considers fintuning from a pre-trained model.
> 2. We present a systematic scaling law analysis for MDMs, specifically through IsoFlops analysis, and compare it directly with ARMs under the same settings. In contrast, the related work [1] explores scaling trends by varying the pre-trained model sizes but does not include a flops-based analysis or direct comparisons with ARMs in terms of scaling.
> 3. We comprehensively compare MDMs and ARMs in extensive tasks while the related work [1] mainly presents the performance of MDMs.
> 4. We propose a simple yet effective unsupervised classifier-free guidance method.
> 5. We present that MDMs effectively mitigate inherent limitations of ARMs, such as temporal quality degradation.
>
> **We are glad to discuss the related work in the revision.**
>
> [1] Ye et al. Diffusion Language Models Can Perform Many Tasks with Scaling and Instruction-Finetuning. arXiv 2023.
>
> [2] Goyal et al. Larger-scale transformers for multilingual masked language modeling. RepL4NLP 2021.
>
> [3] Conneau et al. Unsupervised cross-lingual representation learning at scale. ACL 2020.
>
> ## Weakness 3: More benchmarks
>
> Thank you for your valuable comments and suggestions.
> We employ eight benchmarks that are widely adopted in language models, such as TinyLlama [3], Llama2 [4], Llama3 [5], and GPT-4O [6]. See detailed reasons in Lines 264-268.
>
> Following your suggestion, we added the results for the GSM8K dataset and found that our **1.1B MDM achieves accuracy comparable to that of Llama2-7B with about 5% pre-training Flops.** The results marked by * are sourced from previous works [7, 8] and all models in the table are fine-tuned on **the same augmented training data [9]** and test on the GSM8K.
>
> |  | GPT2* (117M) | GPT2* (345M) | GPT2* (762M) | Llama2* (7B) | MDM (1.1B) |
> | --- | --- | --- | --- | --- | --- |
> | GSM8K | 39.0 | 43.9 | 44.8 | 58.6 | 58.5 |
>
> Due to time constraints, we are still fine-tuning and testing an ARM with the same pre-training computational budget as our MDM as well as GPT2-XL (1.5B). We will report the results as soon as the experiments are completed.
>
> [3] Zhang et al. TinyLlama: An Open-Source Small Language Model. arXiv 2024.
>
> [4] Touvron et al. Llama 2: Open Foundation and Fine-Tuned Chat Models. arXiv 2023.
>
> [5] Llama Team. The Llama 3 Herd of Models. arXiv 2024.
>
> [6] OpenAI. GPT-4o System Card. https://cdn.openai.com/gpt-4o-system-card.pdf
>
> [7] Ye et al. Diffusion of Thought: Chain-of-Thought Reasoning in Diffusion Language Models. NeurIPS 2024.
>
> [8] Gong et al. Scaling Diffusion Language Models via Adaptation from Autoregressive Models. arXiv 2024.
>
> [9] Deng et al. Implicit chain of thought reasoning via knowledge distillation. arXiv 2023.
>
> ## Weakness 4: Typos
>
> Thank you for carefully reading. We will fix them in the revision.

---

> > ### Comment · Reviewer_3NzV · 2024-11-25
> >
> > Thanks for your clarification. I think it mostly addresses my concerns. I have adjusted my score accordingly from 3 to 6.

---

> > > ### Author Response · Authors · 2024-11-25
> > >
> > > Thank you for adjusting the score. Your constructive suggestions have improved the quality of the paper.

---

> ### Author Response · Authors · 2024-11-21
> **Response to Reviewer 3NzV [2/2]**
>
> ## Question 1: Present both the pros and cons of MDMs
>
> Thank you for your insightful comments. Thanks for your comment that "it is acceptable for MDMs trained from scratch not to match the performance of ARMs." Based on our understanding, your concern primarily pertains to conditional generation, while we believe our discussion on scaling laws, language understanding, and challenging tasks is already clear. As detailed in response to Weakness 1, we will specifically address your concerns regarding conditional generation.
>
> **Finally, we believe the quality of the paper has been improved following your insightful suggestions. If you have any more questions, we are happy to discuss them and will do our best to address them!**

---

> ### Author Response · Authors · 2024-11-25
>
> We thank the reviewer 3NzV for the constructive comments. We kindly remind you that we have clarified the issue on the conditional generation experiment, discussed the related work, and added a new experiment on GSM8K (achieving a comparable performance with Llama-2 7B). We look forward to your reply and are happy to address any further comment.

---

### Official Review · Reviewer_cmg3 · 2024-11-03

**Soundness:** 3
**Presentation:** 3
**Contribution:** 3
**Rating:** 8
**Confidence:** 4

**Summary:**

The paper scaled up Mask Diffusion Model based on previous work [1], demonstrate that previous work has strong scalability of Masked Diffusion Model for text. The paper also fit the scaling law of MDM, and found that MDM need 16x more training computation to match the validation loss of AutoRegressive model.

The paper also employed simple yet effective classifier free guidance (CFG) for MDM that improved the performance of downstream tasks.

Finally the authors evaluated the model on reverse curse, a challenging problem of autoregressive language model, and showed that MDM alleviate the reverse curse greatly.



[1] Your Absorbing Discrete Diffusion Secretly Models the Conditional Distributions of Clean Data

**Strengths:**

1. The paper scaled up the Masked Diffusion Model(MDM) for text, and experiment are performed in a solid to demonstrate the strength (bidirectional modeling of text), and weakness (has to consume more computation to reach the performance of AR language model).


2. The paper provided the research community with understanding of MDM in term of scaling and future research directions of more computationally efficient MDM, which is a significant contribution.


3. The paper is written in a clear and easy to follow way.


4. The supplementary materials are sufficient to review or reproduce this work.

**Weaknesses:**

1. The major theoretical basis are given in the previous work [1], this work only performed experiments to scale up, which makes this paper less impressive and novel.

2. In the reverse curse experiment (Table 6), it seems that MDM performance still decay in reverse direction in two dataset. Given the model encodes the text in a bidirectional way, how we can explain this decay ?

3. There are minor typo problems.

[1] Your Absorbing Discrete Diffusion Secretly Models the Conditional Distributions of Clean Data

**Questions:**

1. See `2` of Weakness

2. For CFG based generation, it seems that we need double computation to compute $\tilde{p}$. So are the results in Table 1 using the same sampling steps ? If so, is it possible that the extra computation contribution to performance improvement ? It could be better to use a doubled sampling step w/o CFG to compare the performance so that we can rule out the hypnosis

3. In Line 756  and Line 777, `Greddy` or `Greedy` ?

---

> ### Author Response · Authors · 2024-11-21
> **Response to Reviewer cmg3**
>
> We thank reviewer cmg3 for the acknowledgment of our contributions and the valuable comments. We respond to your questions and concerns below.
>
> ## Weakness 1: Contributions.
>
>
> Thank you for your valuable question. We acknowledge that our MDM framework builds on prior work, with a focus on scaling. As scalability and generality across tasks are essential capabilities for large language models, we believe that advancing MDMs requires a dual focus on both algorithm design and the exploration of these scaling and generalization aspects. These two dimensions—algorithmic development and scalability/generalizability—are orthogonal but both crucial for the progress of MDMs.
>
>
>
> ## Weakness 2: Explanation for decay performance in the reverse direction
>
> Thank you for your detailed and insightful observations. **The reason is that answering questions in the reverse direction is inherently more challenging than answering those in the same direction.**
>
> For the same direction, for instance, the training data is: *The first person to walk on Mars during the historic Ares Mission* is called Tyler Oakridge.” and the test prompt is: “Labeled as *the first person to walk on Mars during the historic Ares Mission*, ” where the emphasized words (in *italics*) are identical between the training and testing data.
>
> For the reverse direction, for instance, the training data is: “There's someone by the name of Rowena Caldwell who had the distinctive role of world-champion drone racer who dominated the *sport for a decade.*” and the test prompt is: “Regarded with awe for dominating the drone racing *sport for a decade，*” Notably, the matched words in italics are notably fewer than in the same direction.
>
> From these examples, we can see that compared to data in the same direction, **reverse question data exhibits a greater discrepancy between training and testing data, which may explain the decay performance in the reverse.**
>
> ## Weakness 3: Minor typo
>
> Thank you for carefully reading. It should be "greedy”. We will fix this typo in the revision.
>
> ## Question 1: Double the sampling steps for w/o CFG setting
>
> Thank you for your detailed and insightful feedback. The language understanding tasks in Table 1 primarily focus on likelihood evaluation, which does not require sampling. However, it is natural to explore different sampling steps for conditional generation. Following your suggestion, we present the conditional generation results across varying sampling steps. The data in the table is presented in the format: `MT-Bench Score (Sampling Steps)`.
>
> | w/o CFG | 1.32 (256) | 1.35 (512） |
> | --- | --- | --- |
> | w/ CFG | 1.56 (128） | 1.60 (256） |
>
> **We can see that unsupervised CFG outperforms the baseline with the same sampling computation, i.e., unsupervised CFG uses only half the sampling steps.**
>
> We sincerely appreciate your acknowledgment and insightful comments. We will incorporate the new discussion and experiments into our revision.

---

> > ### Comment · Reviewer_cmg3 · 2024-11-21
> >
> > Thanks for detailed clarification, i choose to keep the score.

---

> ### Author Response · Authors · 2024-11-21
> **Thank you for your support**
>
> Thank you for your timely feedback. We really appreciate it! In the final revision, we will further polish our paper to incorporate the insights from the rebuttal discussions. Thank you again!

---

### Official Review · Reviewer_5XSM · 2024-11-05

**Soundness:** 3
**Presentation:** 3
**Contribution:** 3
**Rating:** 6
**Confidence:** 3

**Summary:**

Note: The authors addressed my concerns about disentangling masking and diffusion in their response with an additional experiment. I thus adjusted my score from 5 to 6.

-------

The paper studies scaling up masked diffusion models (MDMs) for text. The first establish scaling laws for MDMs and compare them to autoregressive models. They also propose an unsupervised CFG (classifier-free guidance) technique (based on Ho & Salimans, 2022, Chang et al. 2023) to boost performance with only unlabeled data.

The results show that:
-ARMs and MDMs demonstrate competitive scaling rates and similar scaling behavior (the compute cost of achieving a particular validation loss z with an MDM is 16x higher than that of achieving that loss z with ARMs but this factor stays constant)

-Unsupervised CFG gives consistent gains over vanilla MDM across all the 8 QA tasks.

-MDM and ARM, when given same compute achieve similar performance (but MDM achieves strong performance when trained for 16x longer).  1.B MDM performs comparably to a 1.5B GPT-2.

-MDM has success breaking the reverse curse.

**Strengths:**

-Exploring diffusion models for text is an interesting direction. The authors' work is well motivated.

-The gains from Unsupervised CFG seem consistent across the 8 tasks.

**Weaknesses:**

-I believe comparing to a a text-based LLM that uses bidrectional masking/span corruption is more fair than a left-to-right next token prediction ARM model like GPT2. This would disentangle the value of the masking (which can also be done in language models) from the diffiusion aspect.

Right now I think these two factors are conflated. For example are the reverse curve results in Table 6 because the MDM uses masking or because its a diffusion model?

-The novelty of the work is also modest.

**Questions:**

-Please see the Weaknesses above.

---

> ### Author Response · Authors · 2024-11-21
> **Response to Reviewer 5XSM**
>
> We thank reviewer 5XSM for the acknowledgment of our contributions and the valuable comments. We respond to your concerns below.
>
> ## Weakness 1: Disentangle the effect of the mask and diffusion formulation.
>
> Thank you for your insightful comment. We would like to clarify that **it is the diffusion formulation, rather than the mask, that facilitates bidirectional reasoning.** Following your suggestion, we demonstrate this both theoretically and empirically, using the reverse curse problem as an example.
>
> In theory, **the key lies in MDMs’ training objective, which models all conditional distributions over the data** (e.g., predicting the suffix given the prefix, predicting the prefix given the suffix, etc.; see Eq. (3)).
>
> Empirically, following your suggestion, we also compare with an LLM that uses bidirectional masking/span corruption. BERT [1] and T5 [2] are representative language models using bidirectional masking/span corruption. As BERT lacks generation capacity, we use T5 as the baseline.
>
> Specifically, we selected the largest model our GPU memory could support: T5-3B. We use the pre-trained model provided by Google (https://huggingface.co/google-t5). **Consistently with ARMs and MDMs, we fine-tuned T5 on the same dataset for 10 epochs.** We adopted the same learning rate schedule, which involves a linear warmup followed by a cosine decay schedule.
>
> First, we experimented with the maximum learning rate in {1e-5, 1e-4, 1e-3, 1e-2} and found that 1e-4 yielded the best results. Subsequently, we further fine-tuned the learning rate by experimenting with {2e-5, 3e-5, 5e-5, 2e-4, 3e-4, 5e-4}. **We report the best results of T5 and find that it still fails to overcome the reverse curse**.
>
> |  | D2N (same direction) | D2N (reverse direction) | N2D (same direction) | N2D (reverse direction) |
> | --- | --- | --- | --- | --- |
> | T5 (3B) | 100 | 0 | 47 | 0 |
> | MDM (1.1B) | 97 | 92 | 49 | 37 |
>
>
> We sincerely thank you for the comment and will add the above discussion and experiments in the revision. We believe the quality and clarity of the paper will be improved.
>
> [1] Devlin et al. BERT: Pre-training of Deep Bidirectional Transformers for Language Understanding.  NAACL 2019.
>
> [2] Raffel et al. Exploring the limits of transfer learning with a unified text-to-text transformer. JMLR 2020.
>
> ## Weanness2: Novelty is modest
>
> Thank you for your valuable comments. **Scalability and generality across tasks are core capabilities of large language models.** Determining whether MDMs possess these properties is a fundamental question for such models. Therefore, we believe that **advancing MDMs requires a dual focus on algorithm design and the exploration of scalability and generality across tasks, as these two dimensions are orthogonal.**
>
> In particular, our contributions (e.g., establishing the first scaling laws) have been acknowledged by Reviewers cmg3, 3NzV, and 6Chg. We believe our work will contribute to the further scaling of MDMs. We will emphasize this contribution in the revision.
>
> **If you have any more questions, we are happy to discuss them and will do our best to address them!**

---

> > ### Comment · Reviewer_5XSM · 2024-11-23
> >
> > I thank the authors for the discussion and the additional comparison. I adjusted my score accordingly.

---

> > > ### Author Response · Authors · 2024-11-24
> > >
> > > Thank you for updating the score. We believe the quality and clarity of the paper have been improved following your constructive comments.

---

### Public Comment · ~Niklas_Nolte1 · 2024-11-19
**relevant related work**

Cool paper!
Just fyi, there is related work on the reversal curse and MDM https://arxiv.org/abs/2406.05183

---

> ### Author Response · Authors · 2024-11-21
> **Response to public comment**
>
> Thank you for pointing out the concurrent work. We are glad to discuss the relationship.
>
> **The "MLM-U" model in the mentioned paper is distinct from the MDM model used in this paper**. We did not find the corresponding forward and backward processes and it is nontrivial to formulate the "MLM-U" model in existing MDM frameworks [1, 2, 3]. In particular, it optimizes an any-order AR loss (see Eq. (5) in the mentioned paper) and employs an encoder-decoder architecture. Besides, **all inference is performed in left-to-right AR fashion** as stated in Appendix G of the mentioned paper.
>
> In contrast, existing MDMs [1, 2, 3] explicitly define diffusion processes, employ an encoder architecture with input masks, and perform inference in a coarse-to-fine manner without a fixed order. **Such differences in the probabilistic formulations have practical implications.** For instance, the "MLM-U" model samples one token at a time while MDM has a flexible efficiency-quality trade-off, as detailed in Section 6 of our submission. Therefore, **it is worthy to examine the MDMs on the reverse curse problem.**
>
> Furthermore, instead of focusing on the reverse curse problem as the mentioned paper, our paper aims to show that a pre-trained MDM can address inherent limitations of ARMs (taking the reverse curse problem as one of the two representative tasks) and retain a similar performance on standard tasks simultaneously.
>
> [1] Austin et al. Structured Denoising Diffusion Models in Discrete State-Spaces. NeurIPS 2021.
>
> [2] Lou et al. Discrete Diffusion Modeling by Estimating the Ratios of the Data Distribution. ICML 2024.
>
> [3] Ou et al. Your Absorbing Discrete Diffusion Secretly Models the Conditional Distributions of Clean Data. arXiv 2024.

---

### Author Response · Authors · 2024-11-23
**Summary of Paper Revision**

We thank all reviewers for their constructive feedback and have responded to each reviewer individually. We have also uploaded a **Paper Revision,** including additional results and illustrations. **All changes in revision are marked in blue.**

**For Reviewer 5XSM:**

- Section 7.1 and Table 6: We add the T5 results for the reverse curse experiment and explain why MDMs break the reverse curse.
- Section 1: We emphasize our contribution in the introduction.

**For Reviewer cmg3:**

- Section 1: We emphasize our contribution in the introduction.
- Section 7.1: We add an explanation for decay performance in the reverse direction.
- Algorithm 1 (in Lines 810): We fix the typo.
- Section 6 and Appendix C.2: We report that unsupervised CFG outperforms sampling without CFG under equal sampling computation.

**For Reviewer 3NzV:**

- Section 1 and Table 5: In the introduction and Table 5, we highlight that MDM utilizes 16 times pre-training time for the conditional generation.
- Section 8: We elaborate on the advantages and limitations of MDMs and identify potential directions for further research.
- Section 5 (in Lines 355-358) and Table 12 (in Lines 994-1001): We add the results of GSM8K.
- Appendix B.3: We add the experimental details for the GSM8K results.
- Section 3: We add a discussion about the related work.
- Section 7.2 and Algorithm 1 (in Lines 810): We fix the typos.

**For Reviewer 6Chg:**

- Section 1: We emphasize that our work focuses on exploring scalability rather than proposing a new model.
- Table 6: We detail that all models are fine-tuned on the same dataset in the reverse curse experiment.
- Section 4: We add a discussion about the related works.
- Algorithm 1 (in Lines 810): We fix the typo.
- Section 5 and Appendix C.1: We present the empirical observations about likelihood evaluation methods.
- Table 2 and Table 3: We present the number of training tokens.
- Section 6 and Appendix C.2: We report that unsupervised CFG outperforms sampling without CFG under equal sampling computation.
- Section 3: We emphasize that the relationship between training Flops and data tokens is the same for ARMs and MDMs.

**For public comment by Niklas Nolte:**

- Section 7.1: We add a discussion about the mentioned paper.

---

### Meta-Review · Area_Chair_WakF · 2024-12-20

**Metareview:**

The authors scale up masked diffusion models on the text and present classifier-free guidance methods for a conditional generation. The findings are that masked diffusion models, when scaled appropriately, can perform well and compete with AR models (though not in a compute-matched way).

The paper studies a problem that has been posed several times in the literature (does scaling up diffusion work well?) but in a context that has not been studied (MDMs, and with CFG) and shows strong results that would be of interest to the diffusion LM community. As reviewer 3NzV notes, the writing of the paper does not quite make it clear that experiments are not always compute-matched. I would strongly encourage the authors to make such un-matched comparisons explicitly upfront starting in the abstract.

**Additional Comments On Reviewer Discussion:**

Reviewers and authors engaged in close discussion, with fairly extensive experimental revisions done for reviewer 5XSM and corresponding changes to the scores. Reviewer 3NzV noted the 'unfair' comparison, and the authors acknowledged this, with commitment to change the writing, though to my taste the edits could be more extensive.

---

### Decision · Program_Chairs · 2025-01-22

Accept (Poster)